

# Tracking an Atmospheric River in a Warmer Climate: from Water Vapor to Economic Impacts

Francina Dominguez[1], Sandy Dall'erba[1], Shuyi Huang[1], Andre Avelino[1], Ali Mehran[2], Huancui Hu[1], Arthur Schmidt[1], Lawrence Schick[3], and Dennis Lettenmaier[2]

[1]University of Illinois at Urbana-Champaign
[2]University of California Los Angeles, Los Angeles, California, U.S.A.
[3]U.S. Army Corps of Engineers, Seattle District

*Correspondence to:* Francina Dominguez (francina@illinois.edu)

**Abstract.** Atmospheric rivers (ARs) account for more than 75% of heavy precipitation events and nearly all of the extreme flooding events along the Olympic Mountains and western Cascade mountains of western Washington state. In a warmer climate, ARs in this region are projected to become more frequent and intense, primarily due to increases in atmospheric water vapor. However, it is unclear how the changes in water vapor transport will affect regional flooding and associated economic impacts. In this work, we present an integrated modeling system to quantify the atmospheric-hydrologic-hydraulic and economic impacts of the December 2007 AR event that impacted the Chehalis river basin in western Washington. We use the modeling system to project impacts under a hypothetical scenario where the same December 2007 event occurs in a warmer climate. This method allows us to incorporate different types of uncertainty including: a) alternative future radiative forcings, b) different responses of the climate system to future radiative forcings and c) different responses of the surface hydrologic system. In the warming scenario, AR integrated vapor transport increases, however, these changes do not translate into generalized increases in precipitation throughout the basin. The changes in precipitation translate into spatially heterogeneous changes in sub-basin runoff and increased streamflow along the entire Chehalis main stem. Economic losses due to stock damages increased moderately, but losses in terms of business interruption were significant. Our integrated modeling tool provides communities in the Chehalis region with a range of possible future physical and economic impacts associated with AR flooding.

## 1 Introduction

On December 3, 2007, an Atmospheric River (AR) event made landfall on the west coast of the United States. The filamentary plume transporting 1,500 (kg m$^{-1}$ s$^{-1}$) of water vapor at its core extended from the Tropical Pacific, west of Hawaii, to the coast of Oregon and Washington (Fig. 1a). Transformed into equivalent liquid water discharge, this atmospheric river carried approximately 70,000 m$^3$ s$^{-1}$ of liquid water across the 400km cross section of its core, or the about 40 times the average discharge at the mouth of the Mississippi River. Temperatures rose 17°C in less than two hours ahead of the cold front (NOAA, 2008). Along this warm southwesterly tropical air mass, more than 70% of the water vapor and precipitation that reached the coast was of direct tropical origin (Eiras-Barca et al., 2017). This extreme storm with hurricane-force winds was the third in a series of storms that impacted the Chehalis River Basin in western Washington and resulted in six-hour rainfall amounts close





to the 100-year storm volume (NOAA, 2008). The two previous storms (on December 1 and 2) brought heavy snow to the Oregon Coastal Range, the Olympic mountains and the Cascades, while the third and strongest December 3rd event brought mostly liquid precipitation.

The catastrophic flooding along the Chehalis River Basin was primarily due to unusually high and sustained hourly rainfall rates, concentrated in less than twenty-four hours, mainly on December 3. The conditions were exacerbated by warm air advection into the region by the AR, which produced rain on snow conditions, and partially melted the existing shallow, low-elevation snow. Ten US Geological Survey (USGS) stream gauges experienced record flooding, including four on the Chehalis River or its tributaries (Grand Mound, Porter, Doty and South Fork Chehalis; see Fig. 1 for station locations). The peak discharge

measured at Doty was a 500 year event - the only 500 year stream peak event ever recorded in Western Washington. The hurricane force winds, produced wind damage with tree blowdowns, power outages, huge ocean swells and a record coastal storm surge. Eleven people lost their lives. Millions of people lost power throughout the two states as a result of the storm. Portions of interstate 5, the major north-south freight corridor on the West Coast connecting the Puget Sound region of Washington with Oregon and California, were closed for four days resulting in an estimated $47 million in economic losses (WSDOT, 2008).

Major disaster declarations were issued in several counties in the states of Washington and Oregon, but most of the damages were concentrated in three counties in Washington (Grays Harbor, Lewis and Thurston). Lewis County, within which the most affected part of the Chehalis River basin lies, experienced the largest impact with $166 million in damages and 46% of its agricultural land flooded (Lewis County, WA, 2009).

While this event was particularly extreme, more than 50% of the total cool-season precipitation and more than 75% of heavy precipitation (top decile) in the west coast of Oregon and Washington is related to AR events (Rutz et al., 2014). Water vapor transport during the winter season is often roughly orthogonal to the mountain ranges, which favors orographic precipitation enhancement (Neiman et al., 2008; Hu et al., 2017). Furthermore, ARs with anomalous warm, strong low-level water vapor fluxes are responsible for nearly all of the extreme flooding along the Olympic Mountains and the western Cascade Mountains

of Washington (Neiman et al., 2011; Warner et al., 2012; Hu et al., 2017).

    Given the critical role of ARs for precipitation and flooding in the region, it is important to understand how ARs could change in a warmer climate. As tropospheric temperature increases, integrated water vapor transport (IVT) is projected to increase by 30-40% by the end of the $21^{st}$ century along the North Pacific storm tracks, including the west coast of the United States

(Lavers et al., 2015; Salathé et al., 2015). In climate model projections, years with many AR storms are projected to become more frequent, and water vapor content is projected to increase during intense AR events (Dettinger, 2011). The changes in IVT are driven mostly by thermodynamics through increased water vapor content of a warmer atmosphere, while changes in dynamics seem to have only a secondary effect along the northern west coast of the US (Lavers et al., 2015; Salathé et al., 2015; Payne and Magnusdottir, 2015). Based on the analysis of IVT changes, it is tempting to conclude that the projected increase in

intensity and frequency of AR events will lead to increased flooding in the region. However, to quantify the risk of inundation



and its economic impacts, it is important to understand the myriad of processes that happen between the impact of an AR in a watershed and the resulting flooding.

In this work, we present an integrated modeling system that quantifies the atmospheric-hydrologic-hydraulic and economic impacts of the December 2007 AR event (See Fig. 2). In addition, we use the modeling system to project physical and economic impacts under a scenario where the same December 2007 event occurs in an atmosphere with increased greenhouse gas forcing. As opposed to a traditional approach that uses an ensemble of downscaled and bias-corrected climate model simulations, we use the regional model simulations of the December 2007 event in hypothetical future climate settings. We then use these high-resolution simulations in a warmer climate as forcing for the hydrology-hydraulic and economic loss models. Our work follows a similar procedure as the USGS Multihazards Project, which used a synthetic, but plausible, California AR scenario to estimate the human, infrastructure, economic and environmental impacts for emergency-preparedness and flood planning exercises (Porter and andothers, 2010). In our work, we focus on the Chehalis River basin in western Washington to provide an end-to-end model of severe weather, physical impacts and economic consequences of ARs in a warmer climate.

The "pseudo-global warming" (PGW) approach we use, originally proposed by Schär et al. (1996) can provide complementary information to the traditional downscaling approach, with more physical insight into detailed spatial processes and potentially a better way of communicating with regional stakeholders, as argued by Hazeleger et al. (2015). Notably, this method allows us to incorporate different types of uncertainty including: a) alternative future radiative forcings associated with different Coupled Model Intercomparison Project 5 (CMIP5) Representative Concentration Pathways (RCPs) - RCP4.5 and 8.5 scenarios, b) different possible responses of the climate system to future radiative forcings as represented by 14 CMIP5 GCMs and c) different possible responses of the hydrologic system as represented by two different hydrologic models. We do not account for possible changes or structural failures in the main channel hydraulics, and we do not account for possible changes in private/public building infrastructure or trade flows. At each step in the modeling chain we provide an envelope of possible future responses of the system, and present them as changes with respect to the historical control simulation. The modeling system is intended to provide decision-makers with information about the range of physically plausible changes in flood-causing AR storms and floods, as well as a tool to quantify the related economic impacts.

## 2 Data and Methods

The Chehalis River basin, with a drainage area of approximately 5,400 square kilometers, is located in southwest Washington state (Fig. 1b). It heads in the Willapa Hills, flows east, then north and west into Grays Harbor. Most of the basin lies below 1000 m elevation, and fall and winter precipitation mostly occurs as rain, with exceptions in small areas of the extreme northern and eastern portions of the basin. Floods in the basin generally occur in late fall and early winter and are associated with atmospheric



rivers. The most significant floods in the observational period are: Jan 1972, Jan 1990, Nov 1990, Feb 1996, Dec 2007 and Jan 2009 (USGS). In this exercise, we only simulate the Dec 2007 event.

## 2.1 Observations

We used the 1/16° latitude/longitude daily gridded precipitation product derived from NOAA Cooperative Observer (COOP) stations by Livneh et al. (2013). In addition, we used hourly data from seven NOAA (4 COOP and 3 HADS) stations in and around the Chehalis basin (Fig. 1b and Table 1). We used USGS streamflow observations from 15 gauges located throughout the basin (Fig 1b and Table 1). During the flood event, the upstream-most gauge (Doty) measured streamflow up to approximately 60,000 cfs, cut then malfunctioned during the time of peak flood (WSE, 2012), consequently, the peak discharge was estimated by the USGS. In addition, we used the European Centre for Medium-Range Weather Forecasts (ECMWF) Interim Reanalysis (ERA-Interim) (Dee et al., 2011) at 0.75° resolution as lateral boundary conditions for Weather Research and Forecast (WRF) atmospheric model simulations. In terms of economic losses, we rely on HAZUS-MH 3.0 software with its standard infrastructure data and dasymetric dataset for buildings. The 2008 input-output tables, containing information about trade flows across 16 different sectors that represent the economic structure of each of the counties within the state of Washington, were obtained from IMPLAN (2015).

## 2.2 Models

Our atmospheric simulations of the December 2007 event used the Advanced Research version (ARW) of the WRF Model (Skamarock et al., 2005), version 3.4.1, with two nested domains, one of 15km and the inner domain of 3km (Fig. 1a). The time period for our simulation is Nov 30, 2007 to Dec 8, 2007. The physics options used are YSU planetary boundary layer scheme (Hong and Pan, 2009), subgrid-scale convection in the 15km grid based on the Kain-Fritsch parameterization (Kain, 2010), WSM 6-class microphysics (Hong and Lim, 2006) and the Noah-LSM V1.0 (Chen and Dudhia, 2001) land surface model. We tested other microphysics schemes, but we found that WSM 6-class yielded precipitation that was closest to observations.

Our hydrologic simulations used two different models: the U.S. Army Corps of Engineers (USACE) Hydrologic Engineering Center (HEC) Hydrologic Modeling System (HEC-HMS) and the University of Washington's Distributed Hydrology Soil Vegetation Model (DHSVM) hydrologic model (Wigmosta et al., 1994) to estimate the response of the Chehalis watershed to precipitation (Fig. 1b). Our goal in using the two models is to account for uncertainty in the physical representation of hydrologic processes. In HEC-HMS, we partitioned the watershed into 64 sub-basins with homogenous soil and land cover properties based on data from SSURGO (USDA-NRCS) and NLCD 2011 (Homer et al., 2015). HEC-HMS provides the streamflow response of each of the sub-basins that drain to the Chehalis main channel. We calculated baseflow in three different ways: if there was a stream gauge, we used the USGS stream statistics; if the stream gauge was located downstream of a tributary, we calculated the initial base flow for the channel receiving from each sub-basin based on the fraction of the gauged area contributed by each sub-basin in the tributary; if there were no stream gauges available, we estimated the initial base flow through



analogy with similar size sub-basins nearby. We used the Green and Ampt option in HEC-HMS to simulate infiltration in each sub-basin, and calculated the saturated hydraulic conductivity, effective porosity and wetting front suction head based on the hydraulic soil group. For each sub-basin, we used the area-weighted properties. For purposes of calculating soil infiltration rates, we estimated percent impervious area using the land use and land cover maps obtained from SSURGO. The runoff trans-

form uses the Soil Conservation Service (SCS) lag time.

DHSVM is an explicit physically based, spatially distributed hydrological model developed primarily for use in regions with complex terrain. Unlike HEC-HMS, DHSVM uses a rectangular grid formulation, here with a spatial resolution of 150m. DHSVM represents runoff primarily through the saturation excess mechanism, using a representation of a shallow water table

whose depth is modeled similarly to TopModel (Beven and Kirkby, 1979) with the exception that the spatial variation in depth to the water table is represented explicitly, rather than statistically. At each grid cell, unsaturated moisture flow through the root zone is computed using a prescribed hydraulic conductivity which decays exponentially at the water table depth to the saturated hydraulic conductivity. Redistribution of moisture between pixels occurs (only) in the saturated zone, where the hydraulic gradient is take to equal the (computed) slope of the water table, following Wigmosta and Lettenmaier (1999). The model uses

a linear storage scheme to route both overland and subsurface flow (which occurs at the intersection of the water table and the stream network) through a channel network identified using digital topographic data. We calibrated both HEC-HMS and DHSVM using observed daily streamflow at the USGS stream gauges. We calibrated the maximum infiltration rates for each soil type and (for DHSVM) using Manning's coefficients for each channel reach.

We used the output from the two hydrologic models as boundary conditions for the USACE River Analysis System (HEC-RAS) one-dimensional unsteady flow model to perform hydraulic simulations of water levels in the Chehalis River main stem and its largest tributaries. The calibrated HEC-RAS model was provided to our team by USACE. USACE, and its contractor, Watershed Science and Engineering (WSE) updated previously existing hydraulic models of the Chehalis River based on data from a bathymetric survey performed by WSE as well as available LiDAR data. They then calibrated the updated model based

on hydrologic observations in the watershed. The hydraulic model extends from the mouth of the Chehalis River to upstream of Pe Ell (173 km). The model includes portions of the following tributaries: Wynoochee River, Satsop River, Black River, Skookumchuck River, Newaukum River, and South Fork Chehalis (Fig. 1b). HEC-RAS output includes river stage and stream-flow calculations at each channel cross-section, flood inundation extent and flood inundation depth. WSE calibrated the model to the Feb 1996 and Jan 2009 storm events, and used the Dec 2007 storm event for validation. WSE adjusted channel and

overbank values of Manning's $n$ bottom roughness coefficient, flow roughness factors, and the placement of ineffective flow areas in their calibration process. The HEC-RAS model provided by USACE, used observed streamflow hydrographs as lateral boundary conditions; for this reason, we developed our own hydrologic models, as described above to provide flexibility in our simulations of alternative storm scenarios.



We calculated the direct economic losses using HAZUS (HAZard USa), a software developed by the Federal Emergency Management Agency (FEMA, 2015) to calculate economic losses associated to different natural disasters, including floods (see, among others, Ding et al. (2008), Banks et al. (2014), Gutenson et al. (2015)). We used HAZUS-MH version 3.0 and its default dasymetric datasets to calculate how the HEC-RAS-simulated flooding led to direct economic losses to agriculture

(crops), to buildings and public infrastructure such as telecommunication lines and roads. The dasymetric data embedded in HAZUS, which includes information about the location and characteristics of the buildings and infrastructures (e.g. number of floors in a building, number of lanes in a road), allocates the use of land and of buildings by economic sectors so that one can estimate how the direct economic losses result in direct production capacity constraints and losses by sector. Because each company or institution relies on a set of suppliers and purchasers to support its activities, they too will experience production

losses as a result of the flood, even though they have not been flooded themselves. These indirect economic losses are estimated from the 2008 Input-Output tables extracted from IMPLAN at a 16-sector aggregation level (Avelino and Dall'erba, 2016). In addition to production losses, the combination of HAZUS and of input-output techniques allow us to quantify how local final demand decreases as a result of the employees suffering from labor income losses due to temporary closure of their workplace. Reconstruction costs, on the other hand, correspond to a positive stimulus corresponding to the total repair costs of buildings,

infrastructure and vehicles that were destroyed or damaged during the flood. Due to the small size of the economy of the affected counties, the model assumes that reconstruction efforts are supplied by companies located outside of the flooded area. The duration of the recovery phase is given by HAZUS and assumed to be linear in time. The total economic impact in the three affected counties and the rest of Washington is then estimated using the Inventory-Dynamic Inoperability Input-Output Model (Inv-DIIM) proposed by Barker and Santos (2010), that accounts for month-to-month cascading effects on production chains

due to supply restrictions and existing inventories that mitigate some of these losses. In relation to other available input-output models, the Inv-DIIM offers a dynamic view of the inoperability and recovery processes, in addition to accounting for available inventories that can alleviate disruptions in the region (Avelino and Dall'erba, 2016).

## 2.3  Climate Change simulations

To understand how the December 2007 event would change if it occurred in a warmer climate, we used a pseudo-global warm-

ing approach (Schär et al., 1996; Sato et al., 2007; Kawase et al., 2009; Lynn and Druyan, 2009; Rasmussen et al., 2011; Lackmann, 2013, 2015). In this approach, the lateral and initial boundary conditions used in the WRF control simulation are modified by adding a perturbation 'delta' to reflect future changes in temperature, as simulated by Global Climate Model (GCM) projections for the future. We only modified vertical and surface temperature and SSTs, while increasing the specific humidity to maintain constant relative humidity. In this way, we ensured that the storm dynamics remain unchanged (Schär

et al., 1996). It is important to emphasize that this method does not account for possible changes in large-scale dynamics, such as changes in the storm track. However, it has been shown that the changes in future AR events in this region are dominated by thermodynamic (changes in humidity) as opposed to dynamic processes (changes in wind) (Lavers et al., 2015; Salathé et al., 2015; Payne and Magnusdottir, 2015). For this reason, the PGW method provides useful information about possible future AR



changes in the Chehalis basin.

The fourteen different CMIP5 global climate models used to calculate the changes in temperature over the region (WRF model outer domain) are listed in Table 2. Based on one simulation from each model for two different representative con-
centration pathway scenarios (RCP4.5 and RCP8.5) we obtained an envelope of possible changes in temperature between the future (2071-2098) and the historical (1980-2004) mean December-January-February (DJF) temperatures (Fig. 3). We denote 'lower' as the smallest change in temperature and 'upper' the largest. Surface temperature change range between approximately 1 and 4 K, increase to between 2 and 6 K around 350mb and then decreases sharply to approximately -1 to 2 K at 50mb. These patterns are similar to the global-averaged changes in temperature which have maximum warming in the upper troposphere
and cooling in the stratosphere (IPCC, 2013).

We interpolated the domain-averaged changes in temperature from the 'upper' and 'lower' scenarios to the same 26 vertical levels of ERA-Interim. Then, we added these deltas to the ERA-Interim forcing to perform two simulations, one for the 'upper' scenario, and one with the 'lower' scenario. In this way, we are only evaluating the change in precipitation due to horizontally
homogeneous changes in temperature - all other variables remain exactly the same as in the control simulation. This ensures that the AR's path and orientation do not change due to changes in atmospheric dynamics (see mathematical derivation in Schär et al. (1996)).

### 2.4 'Delta Method' for Model Simulations

Each model is sensitive to its input data. In particular, the socioeconomic evaluation requires precise information about the
spatial location and depth of inundation. For this reason, in each part of the model chain, we decided not to use the raw model data but rather the changes in total water flux as simulated by the different models (see Fig. 2). Our strategy for each model simulation was as follows:

1. We performed control simulations of each model forced with observed or reanalysis data (WRF is forced by ERA-Interim, HEC-HMS and DHSVM are driven by observed precipitation, HEC-RAS is forced by observed streamflow).
Due to a lack of observed maximum flood extent, we forced HAZUS with the inundation depth and extent as modeled by the control HEC-RAS simulation.

2. We calibrated each model so as to best simulate the relevant observations.

3. We ran WRF with the PGW conditions, both the 'upper' and 'lower' scenarios, and obtained changes in precipitation (WRF-PGW).

4. Based on the ratio of WRF-PGW and WRF-control precipitation, we obtain a percent change in precipitation over the entire December 1-4 period. We modified the observed precipitation by this percent change, and then ran the hydrologic models with modified precipitation (HEC-HMS-PGW, and DHSVM-PGW).



5. Based on the ratio of HEC-HMS-PGW and HEC-HMS-control (and DHSVM-PGW to DHSVM-control) streamflow,
   for each type of inflow into the main Chehalis channel we obtain a percent change in total streamflow volume for the
   December 1-7 period. We then ran HEC-RAS with modified streamflow (HEC-RAS-PGW).

6. Based on the new HEC-RAS-PGW inundation extent and depth, we run HAZUS and our input-output model to obtained
   new economic loss estimates.

## 3   Historical Simulations

The WRF-control simulation captures the observed extreme precipitation over the Oregon Coastal Range and Olympic Mountains with precipitation on the order of 80 mm day$^{-1}$ over some areas (Fig. 4a and b). However, the simulation overestimates precipitation over the Cascades, and underestimates precipitation over most of the Chehalis basin by about 30-40% (Fig. 4c).

The simulation overestimates precipitation over the Willapa Hills in the southern part of the basin.

HEC-HMS captures the timing of peak stage and flow; however, it has problems with underestimation of peak flow, and more generally underestimates discharge throughout most of the basin (Fig. 5, dashed lines). DHSVM, on the other hand, adequately captures peak flow in the upper basin (Doty and Newaukum), but overestimates peak discharge in the lower basin,

and underestimates recession flows (Fig. 5, dotted lines). As explained above, the hydrologic models are different, and they both have strengths and weaknesses in simulating different parts of the hydrograph at different locations. It is important to note that we used a combination of Livneh precipitation data (daily timescale) with hourly data from five NOAA stations (shown in Fig. 1) used to partition the Livneh daily totals. Hence, while the total daily volumes match the Livneh product, the hourly variability comes from the station data. There is considerable uncertainty in the Livneh precipitation product daily totals for

this storm and even more uncertainty as to the hourly precipitation throughout the basin. Errors in the hydrologic response are largely due to error in the precipitation estimates. Since the 2007 flood, an NWS precipitation radar has been installed (at Langley Hill) and the number of HADS stations has increased, helping to better resolve the space-time distribution of precipitation over the basin. These assets were not, however, available during the 2007 storm.

The calibrated HEC-RAS hydraulic model, driven by observed streamflow from the USGS stations (see station locations in Fig. 1), performs very well (Fig. 6). The differences between observed and simulated stage along the Chehalis main stem range from -0.54 to 0.65 meters, while the difference in peak flow magnitude ranges from about -1.4% at Doty (upstream) to -16.9% at Porter (downstream). The resulting inundation depth and extent are shown in Fig. 6. Large areas around the cities of Chehalis and Centralia (see Fig. 1b for location) were inundated.

We used the inundated areas and depths from HEC-RAS to calculate the local damages to arable land, buildings and content, infrastructure and vehicles using HAZUS and the net loss in local production using the Inv-DIIM. The total physical damages for Lewis, Thurston and Grays Harbor combined were estimated at $678 million with business disruption losses of $51 million





(Table 3, 'Base' rows) most of which was in Lewis County (Avelino and Dall'erba, 2016). While reported loss estimates are difficult to obtain, the Department of Commerce estimated losses for the states of Washington and Oregon combined for this flooding event were approximately $1 billion dollars, so our estimates for the three counties seem reasonable. In addition, the official building and inventory damages in Lewis county were estimated at $166 million (Lewis County, WA, 2009), close to our estimate of $151 million for the same categories.

It is important to re-iterate at this point that we do not use the raw model output (from WRF, HEC-HMS, DHSVM or HEC-RAS) to drive the subsequent model in the historical simulations (Fig. 2). The individual historical model simulations are used as the 'control' to compare with the climate change simulations, described below.

## 4    Climate Change Simulations

In the WRF-PWG simulation, we added the changes in temperature shown in Fig. 3 (both 'upper' and 'lower' scenarios) to each level of the ERA-Interim boundary conditions used in the control simulation, while maintaining constant relative humidity. This necessarily implies an increase in the specific humidity, as higher temperatures increase the saturation specific humidity. These changes induce variations in the IVT of the projected AR event, which increases by 12.6% in the lower scenario to 38.5% in the upper scenario for the WRF outer domain (Fig. 7 shows the spatial changes for the PGW-upper scenario). The increase approximately follows the Clausius-Clapeyron scaling of about 7% per degree of warming. The increase in IVT can be as large as 500 kg m$^{-1}$ s$^{-1}$ throughout the AR corridor. IVT also increases within the inner WRF domain by 12.4% to 42.3% for the two scenarios. Water vapor mixing ratio increases everywhere, but not homogeneously in space (Fig. 8), with a clear structure of changes above 40% at the 800mb level. However, due to the differences in temperature, the relative humidity can increase or decrease in the PGW-upper simulation, and this leads to both positive and negative changes in cloud water mixing ratio. In Fig. 8 we show these results at the 800mb level, but these heterogenous changes in relative humidity and cloud water can be seen throughout the lower troposphere. As a consequence, precipitation shows both areas of significant increase and decrease throughout the WRF inner domain. The inner-domain area-averaged precipitation change is 8.2% for the lower scenario and 17.8% for the upper scenario - significantly below the Clausius-Clapeyron scaling. At the basin scale, precipitation increases significantly (exceeding 30%) in the northern part of the watershed, and deceases significantly (below 30%) in the southeastern Chehalis basin. We calculated the fractional changes in precipitation for each sub-watershed as the total precipitation that accumulated between Dec 1-4 of the WRF-PGW simulation divided by the WRF-control accumulated precipitation for the same period (Fig. 9a). The upper basin (lowest sub-basin numbers) clearly show precipitation increases, the eastern part of the basin shows decreased precipitation, and the lower basin shows increased precipitation.

We multiplied the observed precipitation by the fractional change in precipitation (shown in Fig. 9a for each HEC-HMS sub-basin), and used the result to force the HEC-HMS and DHSVM PGW simulations. There are two different scenarios, which result in four different hydrologic simulations (HEC-HMS-lower, HEC-HMS-upper, DHSVM-lower and DHSVM-upper). The



results show that some regions generate significantly more runoff due to increased precipitation, while the southeastern part of the basin generates less runoff (Fig. 9b and Fig. 10). Notably, the Doty station in the headwaters of the basin, shows an increase in peak runoff that ranges from 13% in DHSVM-lower to 44% in HEC-HMS-upper. The use of the two hydrologic models provides an envelope of uncertainty in the numerical representation of the hydrologic response (Fig. 10). We find that
5  the sharp increase in streamflow in the headwaters dominates the response in the main channel (in all but the DHSVM-lower), as simulated by HEC-RAS (Fig. 11). There is an increase in both stage and flow throughout most of the channel, with increases that range from about 12-42% in the headwaters (depending on the scenario), to -6 to 5% in the eastern part of the basin, and then about 10-30% at the outlet into Grays Harbor (Fig. 11 and 12a). Again, only the DHSVM-lower scenario shows small decreases in the eastern part of the basin. Despite significant increases in streamflow, the changes in inundation extent are min-
10 imal (Fig. 12b). The reason for this result is that the December 2007 event was so intense that the flooding extended throughout much of the flood plain to the bounding and steeper hills. The PGW simulated increased flood depths, but not much change in flood extent. It is important to reiterate that we did not simulate the failure of any existing hydraulic structures.

The associated socioeconomic losses, as simulated by HAZUS and Inv-DIIM, show an increase in physical damages of
15 2-33% in Grays Harbor County, 9-171% in Lewis County and -1-10% in Thurston County (depending on the scenario and the hydrologic model used) (Table 3). Interestingly, in terms of business interruption losses, the increases are substantially higher and can be very different from the changes in physical damages (27-250%, 14-314% and 46-619% respectively). The flow losses increase dramatically in the DHSVM simulations because of the sectors that are affected during inundation. For example, in Thurston County, public sector and services are significantly affected in the DHSVM simulation, and because
20 these sectors have the largest production and linkages in the county, the impacts increased significantly (480-619%). This indicates that, depending on the hydrological impacts, the simulated economic scenarios can lead to flooding patterns that impact key interconnected sectors in these regions, significantly increasing negative spillover effects. Moreover, due to stronger production-chains in Thurston than in Lewis and Grays Harbor, despite the modest increase in damages, business losses surge the most in Thurston (Table 3). Interestingly, the economy outside of these three counties is positively impacted as reconstruc-
25 tion and recovery efforts stimulate production in the rest of Washington. As a result, the net impact on local production and trade is positive.

## 5  Conclusions

ARs are responsible for most of the extreme winter flooding events in the western United States. As the climate warms, the thermodynamic response of these atmospheric structures will likely lead to significantly more water vapor content and fluxes.
30 Others have hypothesized that a warmer climate will lead to more intense AR-related flooding events and societal impacts. However, the way that the water vapor carried by an AR is transformed into precipitation, runoff, and streamflow along a channel is highly nonlinear and depends on a myriad of fine-scale processes both in the atmosphere and the land surface. Furthermore, the economic impacts depend on both the human footprint, economic structures in the affected areas and trade





linkages with other regions. Because of the risk associated to these events, we need appropriate tools to assess the socioeconomic impact of ARs in a warmer climate.

We have presented an integrated modeling tool that tracks an AR - from its atmospheric development to the economic impacts related to inundation and flooding. We have used this tool to understand how the ARs and their impact could change in a warmer climate using a PGW approach. As argued by Hazeleger et al. (2015), this type of approach is particularly useful for the affected communities because it uses high-resolution models to simulate an extreme hydrologic event that occurred in the past, which the community can remember. The method is flexible enough to tailor the projections to a narrative; in this case 'how would this extreme event change in a warmer climate?'. Furthermore, the method takes into account three types of uncertainty: a) uncertainty if future radiative forcing, b) uncertainty in the climate system response to this radiative forcing, c) uncertainty in the hydrologic response of the system. In this way providing the community with a range of uncertainty of possible future conditions.

In our application to the December 2007 AR-flooding event over the Chehalis river basin, we found that while there is a clear intensification of AR specific humidity and integrated vapor transport for both the 'lower' and 'upper' PGW scenarios, these changes do not translate into generalized increases in precipitation throughout the basin due to spatially heterogeneous changes in relative humidity and water vapor mixing ratio. For this reason, some parts of the basin receive more precipitation, while others receive less. These changes in precipitation translate into amplified changes in sub-basin runoff (in terms of percent change in water mass). But, because the upper basin runoff increases substantially, the streamflow along most of the Chehalis main stem increases in the warming scenarios. Interestingly, this event was so large, that even in the control simulation most of the inundated area was occupied. As a consequence, while the PGW simulation resulted in significant changes in inundation depth, changes in the inundated area were minor. However, these changes in flood depth resulted in economic losses due to stock damages that ranged between -1% and 171%, while losses in local production and trade within the three impacted counties was between 14% and 619% (depending on the affected county, PGW scenario and hydrologic model). The economy outside of these three counties actually benefitted from reconstruction efforts after the flood.

The meteorology, hydrology combined with public policy and mitigation cost-benefits considerations will remain a difficult challenge in the future for the Chehalis Basin. Flooding potential may need to be re-considered in light of possible changes in atmospheric rivers in a warmer climate. Our integrated modeling tool provides communities in the Chehalis region with a range of possible future physical and socioeconomic impacts associated to AR flooding. The framework takes into consideration several important sources of uncertainty. It can be applied to other intense flooding events that perhaps affected other parts of the basin. Furthermore, the tool can be modified to understand different future scenarios, including failure of hydraulic structures, changes in land use/land cover etc. In this way, communities in the region will be better prepared to mitigate the losses and improve disaster relief efforts associated to likely changes in precipitation and flooding that a warmer climate will bring.



*Acknowledgements.* Support for this study has been provided in part by the National Aeronautics and Space Administration (NASA) Grant NNX14AD77G. Any opinions, findings, and conclusions or recommendations expressed in this publication are those of the authors and do not necessarily reflect the views of NASA.



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



**Table 1.** Streamflow and Precipitation Observations. Map ID corresponds to the locations on the map of Fig. 1b.

| Map ID | ID | LON | LAT | Location |
|---|---|---|---|---|
| Streamflow | | | | |
| 1 | 12020000 | -123.28 | 46.62 | Doty |
| 2 | 12020800 | -123.08 | 46.45 | South Fork |
| 3 | 12024400 | -122.77 | 46.67 | Newaukum |
| 4 | 12024000 | -122.68 | 46.58 | Newaukum |
| 5 | 12025100 | -122.98 | 46.66 | Chehalis |
| 6 | 12025700 | -122.59 | 46.77 | Centralia |
| 7 | 12026150 | -122.74 | 46.79 | Skookumchuck |
| 8 | 12026400 | -122.92 | 46.77 | Skookumchuck |
| 9 | 12027500 | -123.03 | 46.78 | Grand Mound |
| 10 | 12031000 | -123.31 | 46.94 | Porter |
| 11 | 12035000 | -123.49 | 47.00 | Satsop |
| 12 | 12035100 | -123.60 | 46.96 | Montesano |
| 13 | 12035400 | -123.61 | 47.38 | Wynoochee Grisdale |
| 14 | 12036000 | -123.65 | 47.30 | Wynoochee Aberdeen |
| 15 | 12037400 | -123.65 | 47.01 | Wynoochee Montesano |
| Precipitation | | | | |
| a | 456864 | -123.85 | 47.475 | |
| b | 451934 | -123.22 | 47.424 | |
| c | 456114 | -122.903 | 46.973 | |
| d | 452984 | -123.504 | 46.543 | |
| e | | -123.083 | 46.343 | |
| f | | -122.908 | 46.61 | |
| g | | -122.458 | 46.596 | |



**Table 2.** CMIP5 GCM models used in this study, including the respective RCP scenario used.

| Model | Institution | Reference | Scenario (RCP) |
|---|---|---|---|
| BCC-CSM1.1 | Beijing Climate Center, China Meteorological Administration, Chin a | Xiao-Ge et al. (2013) | 8.5 |
| CanESM2 | Canadian Centre for Climate Modelling and Analysis, Canada | Arora et al. (2011) | 4.5, 8,5 |
| CCSM4 | National Center for Atmospheric Research, United States | Gent et al. (2011) | 4.5,6.0,8.5 |
| CNRM-CM5 | Centre National de Recherches Meteorologiques / Centre Europeen de Recherche et Formation Avancees en Calcul Scientifique, France | Voldoire et al. (2013) | 4.5, 8,5 |
| CSIRO-Mk3.6.0 | Commonwealth Scientific and Industrial Research Organisation in collaboration with the Queensland Climate Change Centre of Excellence, Australia | Rotstayn et al. (2010) | 4.5, 8.5 |
| INM-CM4 | Institute for Numerical Mathematics, Russia | Volodin et al. (2010) | 8.5 |
| IPSL-CM5A-LR | Institut Pierre-Simon Laplace, France | Dufresne et al. (2013) | 4.5,6.0,8.5 |
| MIROC5 | Atmosphere and Ocean Research Institute (The University of Tokyo), National Institute for Environmental Studies, and Japan Agency for Marine-Earth Science and Technology | Watanabe et al. (2010) | 4.5,6.0,8.5 |
| MIROC-ESM | Japan Agency for Marine-Earth Science and Technology, Atmosphere and Ocean Research Institute (The University of Tokyo), and National Institute for Environmental Studies | Watanabe et al. (2010) | 4.5,6.0,8.5 |
| MPI-ESM-LR | Max Planck Institute for Meteorology (MPI-M), Germany | Zanchettin et al. (2013) | 4.5, 8,5 |
| NorESM1-M | Norwegian Climate Centre, Norway | Zhang and Yan (2012) | 4.5,6.0,8.5 |
| GFDL-CM3 | Geophysical Fluid Dynamics Laboratory, United States | Donner et al. (2011) | 4.5, 8,5 |
| GFDL-ESM2M | Geophysical Fluid Dynamics Laboratory, United States | Donner et al. (2011) | 4.5,6.0,8.5 |
| HadGEM2-ES | Met Office Hadley Centre , United Kingdom | Jones et al. (2011) | 4.5,6.0,8.5 |





**Table 3.** Projected economic losses for the historical simulations (Base) and the upper and lower scenarios for the two hydrologic models. Values are in millions of 2008 U.S. dollars.

**Stock Damages (Private and Public buildings, Content and Inventory; Infrastructure; Vehicles**

| | Grays Harbor | | Lewis | | Thurston | | Rest of WA | | Total Impact | |
|---|---|---|---|---|---|---|---|---|---|---|
| Base (USACE) | $(177) | | $(425) | | $(76) | | | | $(678) | |
| Lower Bound (HEC-HMS) | $(180) | 2% | $(462) | 9% | $(82) | 8% | $- | | $(724) | 7% |
| Lower Bound (DHSVM) | $(218) | 23% | $(1,006) | 137% | $(75) | -1% | $- | | $(1,299) | 92% |
| Upper Bound (HEC-HMS) | $(191) | 8% | $(472) | 11% | $(79) | 4% | $- | | $(743) | 10% |
| Upper Bound (DHSVM) | $(235) | 33% | $(1,151) | 171% | $(84) | 10% | $- | | $(1,470) | 117% |

**Net Impact in Local Production and Trade (Flow Losses)**

| | Grays Harbor | | Lewis | | Thurston | | Rest of WA | | Total Impact | |
|---|---|---|---|---|---|---|---|---|---|---|
| Base (USACE) | $(8) | | $(38) | | $(5) | | $954 | | $903 | |
| Lower Bound (HEC-HMS) | $(10) | 27% | $(44) | 14% | $(7) | 46% | $1,019 | 7% | $958 | 6% |
| Lower Bound (DHSVM) | $(20) | 161% | $(144) | 277% | $(29) | 480% | $1,829 | 92% | $1,636 | 81% |
| Upper Bound (HEC-HMS) | $(11) | 45% | $(45) | 17% | $(8) | 51% | $1,045 | 10% | $982 | 9% |
| Upper Bound (DHSVM) | $(27) | 250% | $(158) | 314% | $(36) | 619% | $2,070 | 117% | $1,849 | 105% |

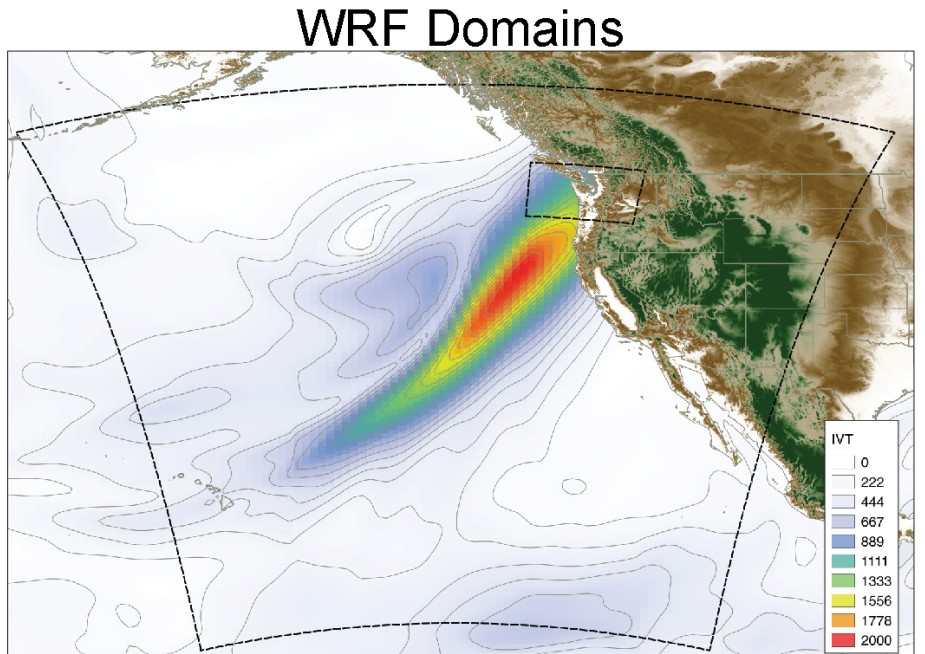

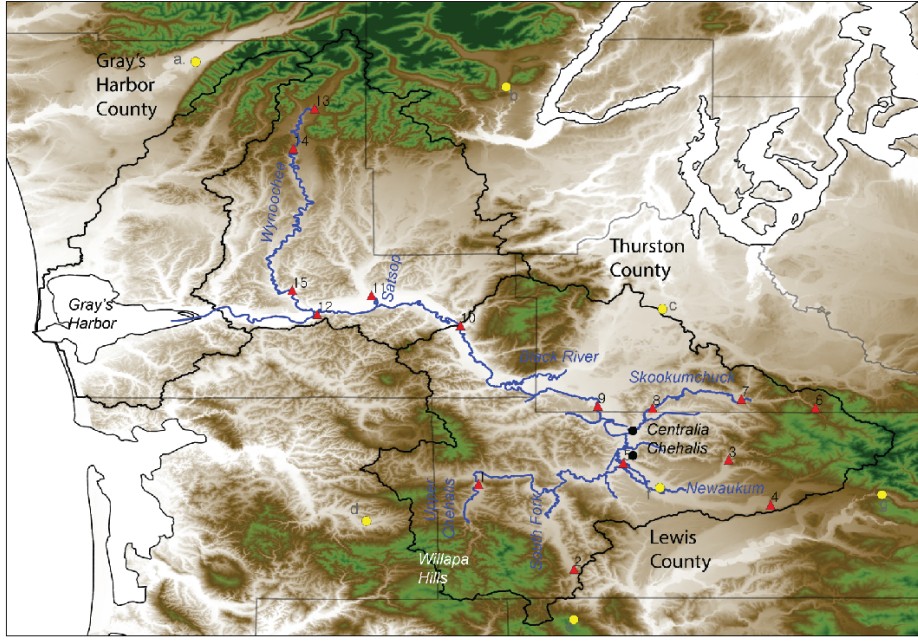

**Figure 1.** (Top) Integrated Vapor Transport (IVT) kg m-1 s-1 on December 3, 2007 from ERA-Interim Reanalysis, dashed lines are the WRF outer and inner domain. (Bottom) Chehalis river basin with topographical features and the largest urban areas (Centralia and Chehalis). The Chehalis main channel as represented in HEC-RAS is shown, along with the USGS gauging stations (red triangles) and precipitation stations (yellow circles) used in this study. Numbers correspond to the station information in Table 1.



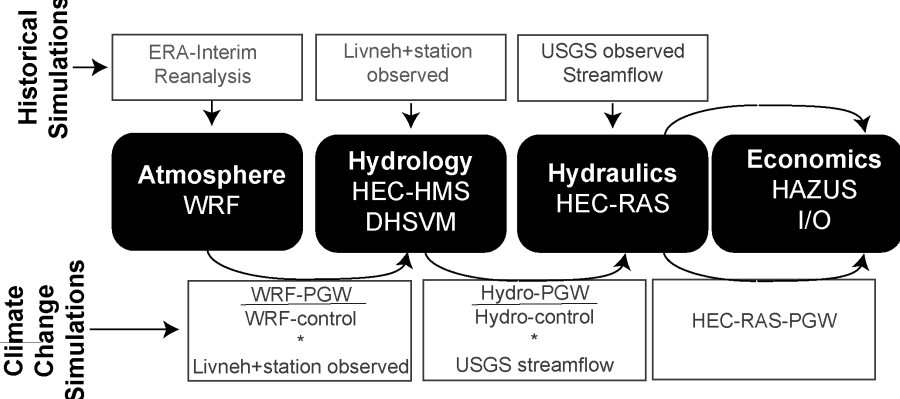

**Figure 2.** Diagram of the integrated modeling, including the models used and the input data for each model during the historical simulations (top) and the climate change simulations (bottom). Hydro-control represents both HEC-HMS and DHSVM-control simulations, while Hydro-PGW represents both HEC-HMS and DHSVM-PGW simulations.





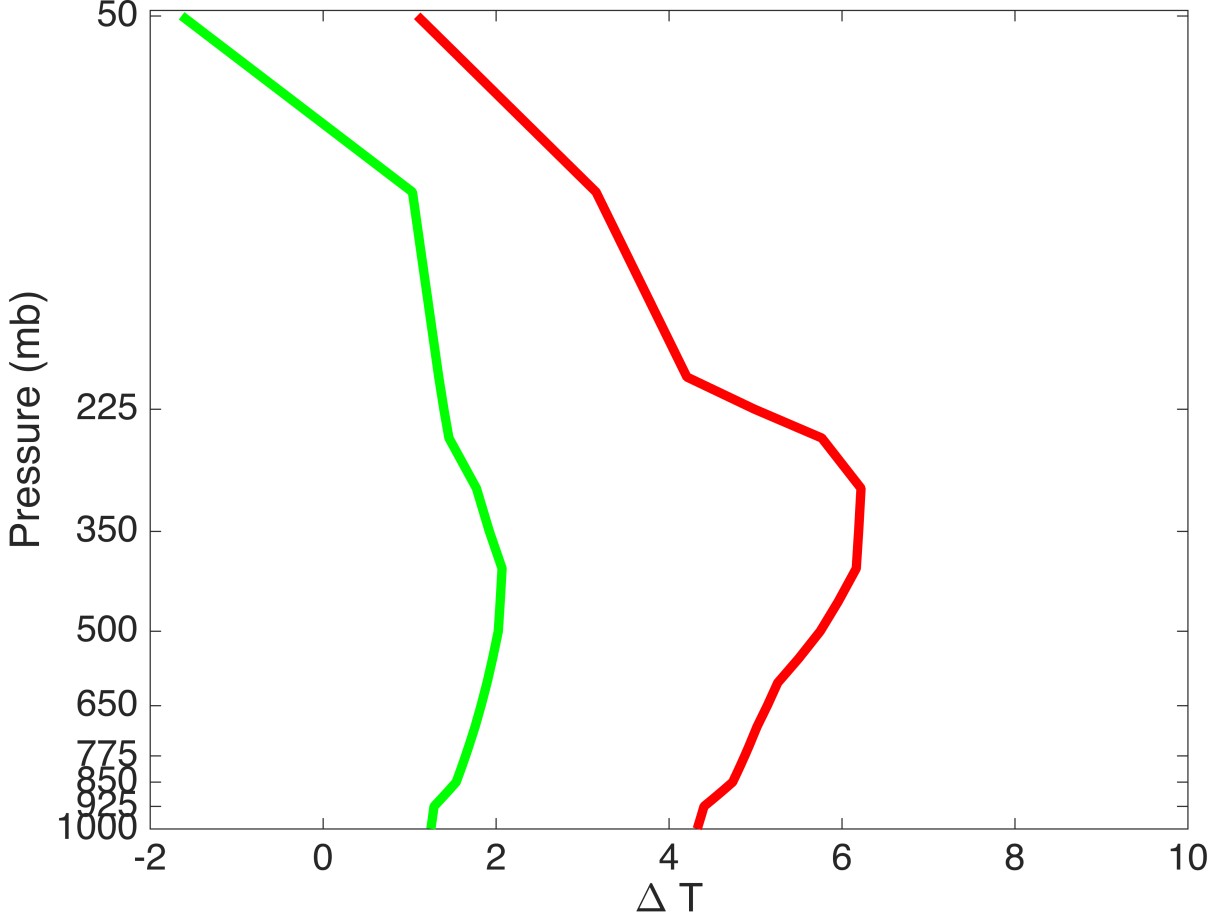

**Figure 3.** Upper (red) and lower (green) bounds of the area-averaged temperature changes as represented by the 14 CMIP5 models listed in Table 12, using the RCP4.5 and RCP8.5 simulations.



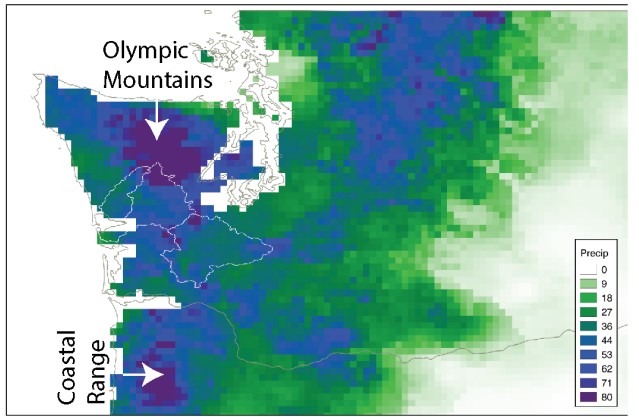

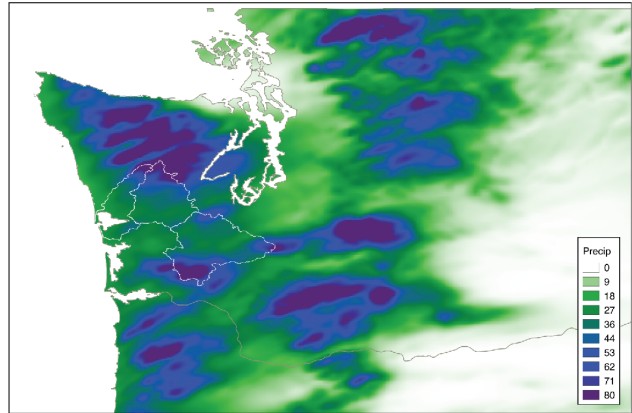

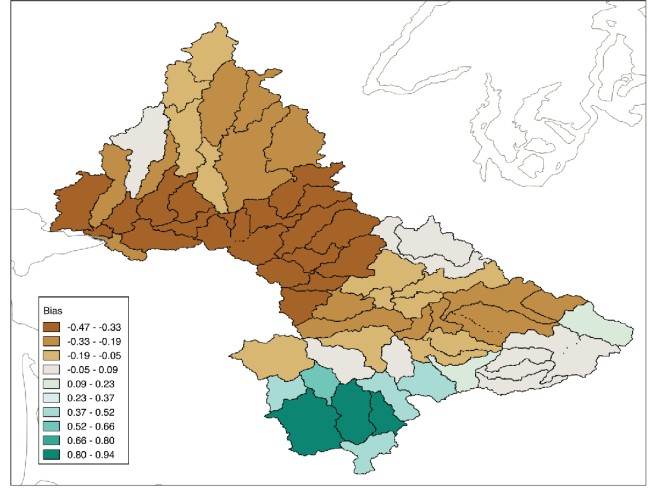

**Figure 4.** (a) Observed daily precipitation (mm day$^{-1}$) averaged for Dec 1-4 of 2007 from Livneh et al. (2013), b) WRF-control simulated precipitation for the same period and c) bias in simulated precipitation for each of the HEC-HMS sub-basins within the Chehalis basin.





**Figure 5.** USGS observed (solid black), HEC-HMS simulated control (dashed blue) and DHSVM simulated (dotted blue) discharge for 4 representative sub-basins within the Chehalis.



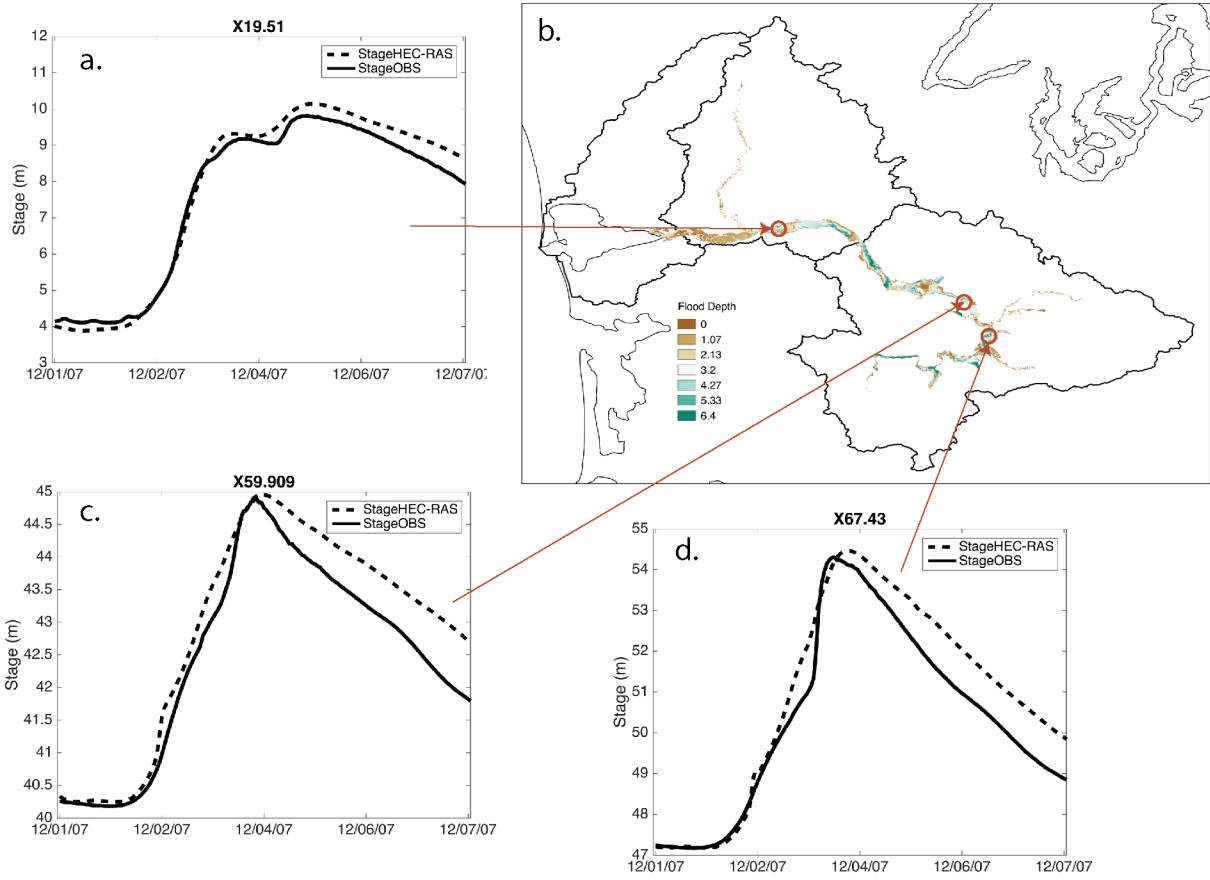

**Figure 6.** (a., c. and d.) USGS observed (solid) and simulated (dashed) stage for three cross-sections of the Chehalis river main stem as represented by HEC-RAS. b. Flood extent and depth map as simulated by HEC-RAS.





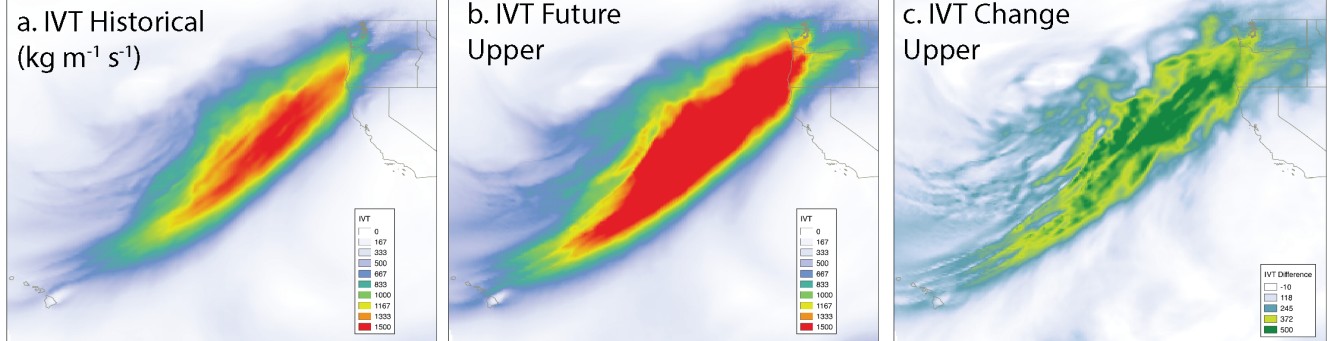

**Figure 7.** a) WRF-control simulated IVT (kg m-1 s-1) for December 3, 2007, b) WRF-PGW simulated IVT (kg m-1 s-1) for December 3, 2007 for the upper scenario, c) Absolute change in IVT, between WRF-PGW upper scenario and WRF-control.



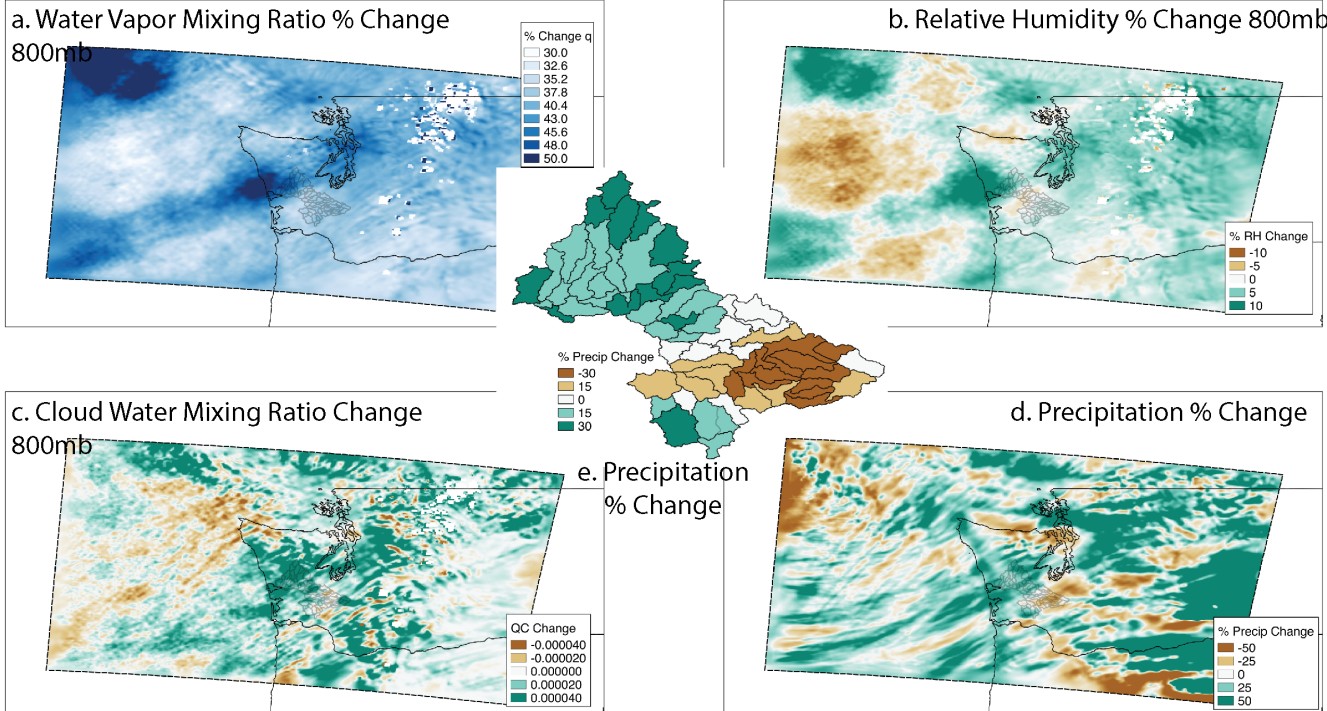

**Figure 8.** (Changes between WRF-PGW for the upper scenario and WRF-control (inner WRF domain), averaged for the Dec 1-4 period for a) water vapor mixing ratio percent change at 800mb, b) relative humidty % change at 800mb, c) absolute change in cloud water mixing ratio at 800mb and d) % change in precipitation e) % precipitation change area averaged over all Chehalis sub-basins of the HEC-HMS model.





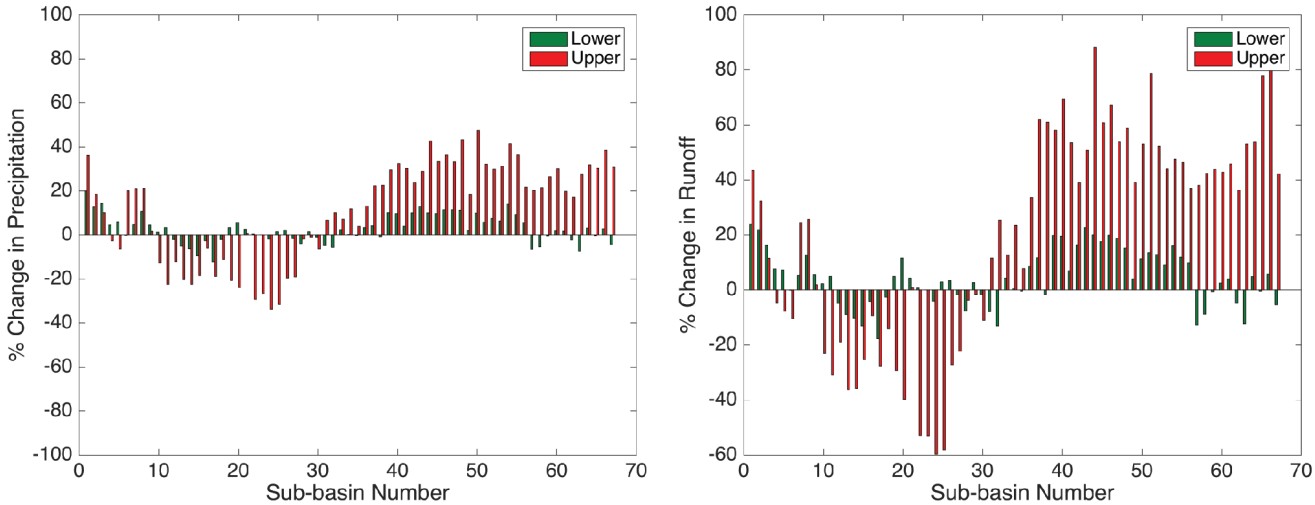

**Figure 9.** (Left) % change in precipitation for the lower (green) and upper (red) scenarios, as simulated by WRF for all Chehalis sub-basins used in the HEC-HMS simulations. (Right) % change in streamflow for the lower (green) and upper (red) scenarios, as simulated by HEC-HMS for all Chehalis sub-basins







**Figure 10.** Streamflow hydrographs for HEC-HMS-PGW upper scenario (dashed) and HEC-HMS control (solid) for select sub-basins in the Chehalis.





**Figure 11.** Streamflow hydrographs for observed (black) and simulated using HEC-HMS (dashed) and DHSVM (dotted) for the lower scenario (green) and higher scenario (red).



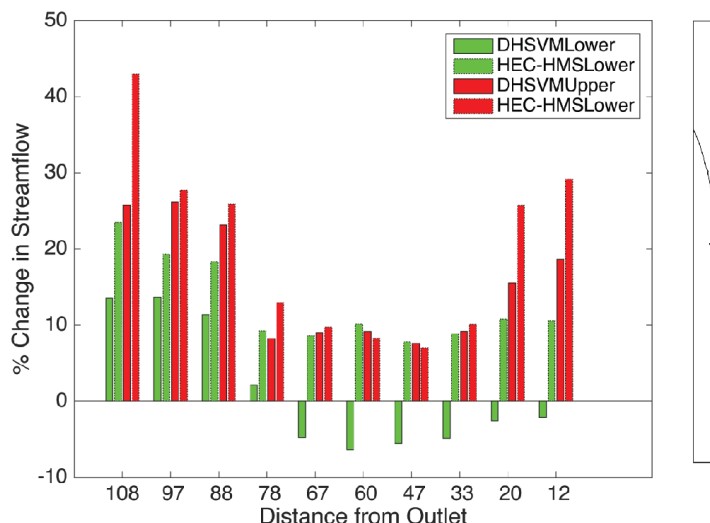
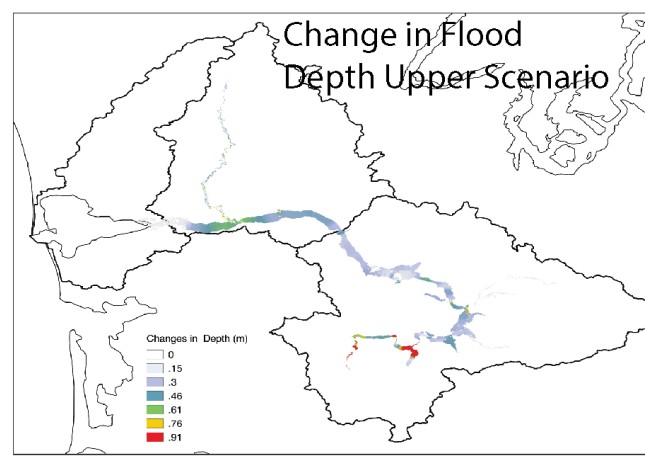

**Figure 12.** Change in streamflow and flood depth along main channel.