# Peer review of "Tracking an Atmospheric River in a Warmer Climate: from Water Vapor to Economic Impacts"

_Earth System Dynamics, 2017_

## Referee Comment (RC1) · Anonymous Referee #1 · 28 Jul 2017

The paper is mainly concerned with a heavy precipitation (atmospheric river) event which occurred around the Chehalis river basin in December 2007. The goal of the paper is to simulate how climate change might affect this kind of event, including economic impacts. In the introduction, the actual event and its properties are presented compactly. The need for research regarding possible changes of atmospheric rivers due to climate change is convincingly motivated. The idea is to develop a coupled hydrologic-hydraulic-economic simulation model. Climate change is represented by the concept of "pseudo-global warming". Section 2 discusses the employed data and models. Main models are WRF (ARW) for the atmosphere, HEC-HMS and DHSVM for hydrology, the HAZUS model for direct economic losses and IMPLAN input-output table for the indirect (or induced) losses. The third section considers the simulation of

the actual event. The fourth section concerns the simulation of the effect of climate change on the considered event.

Comments:

Some parts of the introduction seem to go very much into detail. For example, the event is discussed with its detailed properties, but the region as such is only introduced in section 2. The introduction could be shorter and more general. All specific information could be moved to subsequent sections. For example, the concept of "pseudo-global warming" does not become clear from what is written in the introduction anyhow. The structure could be improved elsewhere. For example, climate change is the topic of section 2.3 as well as section 4. I suggest integrating section 2.3 into section 4. The third section considers the simulation of the actual event, or the model calibration, as I would name it. However, it remains a bit unclear how well the overall model fits the observed data. The fit of some submodels (for precipitation, discharge flows) seems to vary a lot by time, location and so on (e.g. Figure 5). Regarding the economic submodel, detailed economic losses seem to be unknown (p. 9, top), so I believe that HAZUS and the input-output model were not in fact "calibrated" to the event. The authors could be clearer about this. Most importantly, one would expect a summary regarding the authors' judgement of the OVERALL model performance in replicating the historical data.

I can comment mainly on the economic aspects. The general idea of calculating direct losses first and then using an input-output model to calculate indirect or induced losses is plausible. The assumption that reconstruction is done by companies outside the affected area is also common. Regarding the obtained economic figures for the effect of climate change, they seem rather inconclusive. For example, what does it imply that physical damages of the considered event increase between 9 and 171% in Lewis County? The most relevant economic figure (for households, policy makers, insurance companies) would most likely be the expected annual losses and how these are affected by climate change. In particular, the probability of occurrence of the December

2007 event under present and future climate would be relevant in that regard. If we are talking about a 500-year event (as indicated on p. 2), future changes in this particular event would probably not be too relevant. Therefore, I wonder whether it would be possible to calculate hypothetical losses for, e.g. 20-, 50- and 100-year events. The meteorological records should provide the corresponding amounts of precipitation for these events and the economic losses could be obtained by using the model with the calibration for the December 2007 event. The PGW approach (as far as I understand it) would be applicable to those more frequent events analogously. Eventually, the expected annual losses (now and under climate change) could be calculated (see e.g. Velasco, 2015 for a simple approach).

Conclusion:

The overall quality of the paper is good and the suggested revision is somewhere between major and minor. The topic of the paper is relevant and the development of a coupled hydrologic, hydraulic and economic model is plausibly presented. The structure of the paper could be still improved and the implications of the results should be presented more clearly.

Detailed aspects:

The abstract is very long (250 words). I would suggest leaving out the first three sentences, and starting the abstract with "In this work. . .".

Literature:

Velasco, M. et al (2015): Flood damage assessment in urban areas. Application to the Raval district of Barcelona using synthetic depth damage curves.

---

## Referee Comment (RC2) · Anonymous Referee #2 · 16 Aug 2017

1. Does the paper address relevant scientific questions within the scope of ESD?

This manuscript focuses on a single weather event and its hydraulic and economic impact. The multi-model methods are then applied to RCP8.5 and RCP4.5 scenarios. The primary scientific questions presented are (1) what types of uncertainty exist and how are they quantified? (2) which type of uncertainty is most important for quantifying impacts on humans, infrastructure and economics? This in turn is applied to future scenarios.

2. Does the paper present novel concepts, ideas, tools, or data?

The uniqueness of this study is the attempt to provide a full system analysis of an Atmospheric River (AR) event, the hydrologic surface response and hydraulic chan-

nel and overland flow response and the economic costs associated with flooding and infrastructure damage.

The pseudo global warming (PGW) methodology is is novel and computational somewhat efficient, allowing for a zoomed in study with full dynamics.

3. Are substantial conclusions reached?

The analysis provides economic loss previously not performed in such a fashion.

4. Are the scientific methods and assumptions valid and clearly outlined?

Most of the analysis is well done. However, there are statements regarding the amount of water associated with the AR that is not backed up by any analysis. Are the authors confident the AR plume carried 1,500 kg/(ms) and the amount of precipitation that reached the basin was 70,000 CMS?

5. Are the results sufficient to support the interpretations and conclusions?

Yes.

6. Is the description of experiments and calculations sufficiently complete and precise to allow their reproduction by fellow scientists (traceability of results)?

Yes. The descriptions of the models are well done. Error analysis of WRF precipitation is well explained, however hydrologic models calibration and verification may not be sufficiently presented.

Climate change simulations are clear and sufficient.

The HAZUS economic model is unclear with regard to the assumptions and uncertainties. The description of its setup, calibration and verification need to be further explained.

7. Do the authors give proper credit to related work and clearly indicate their own new/original contribution?

Yes.

8. Does the title clearly reflect the contents of the paper?

The title is general with regard to the impacts of ARs and would be more clear if this was presented as a case study based on the December 2007 and RCP85 and RCP45 scenarios on Chehalis River Basin.

9. Does the abstract provide a concise and complete summary? Yes.

10. Is the overall presentation well structured and clear?

Yes.

11. Is the language fluent and precise?

12. Yes.

13. Are mathematical formulae, symbols, abbreviations, and units correctly defined and used?

Yes.

14. Should any parts of the paper (text, formulae, figures, tables) be clarified, reduced, combined, or eliminated?

15. Are the number and quality of references appropriate? Yes.

16. Is the amount and quality of supplementary material appropriate? No supplementary material provided.

---

## Author Comment (AC1) · 13 Sep 2017

**Responses to Reviewer 1**

We thank Reviewer 1 for the thoughtful comments. We have responded to each comment below. We believe that these modifications will significantly improve the manuscript.

*Some parts of the introduction seem to go very much into detail. For example, the event is discussed with its detailed properties, but the region as such is only introduced in section 2. The introduction could be shorter and more general. All specific information could be moved to subsequent sections. For example, the concept of "pseudo-global warming" does not become clear from what is written in the introduction anyhow. The structure could be improved elsewhere. For example, climate change is the topic of section 2.3 as well as section 4. I suggest integrating section 2.3 into section 4.*

We agree with the Reviewer's comments. The details of the 2007 event will be moved to Section 2 – Data and Methods.
The sentences that provide details about the "pseudo-global warming" method will be shifted to Section 2.3 – Climate Change Simulations
In this way, the introduction will be shorter and more general. It will also be clearer that we want to emphasize the new tool, as opposed to focusing on this case study.

We have decided not to integrate section 2.3 into 4. The reason is that section 2.3 is in the "Methods" section, where we go into the details of the PGW methodology. However, section 4 is part of results, so we think it would be confusing for the reader. We will change the titles of each section to clarify each section:

 1 Introduction
2 Data and Methods
      2.1 Data: Observations
      2.2 Methods: Models
      2.3 Methods: Climate Change simulations
         'Delta Method' for Model Simulations goes here but is not labeled as a separate
         section.
3 Results: Historical Simulations
4 Results: Climate Change Simulations
5 Conclusions

*The third section considers the simulation of the actual event, or the model calibration, as I would name it. However, it remains a bit unclear how well the overall model fits the observed data. The fit of some submodels (for precipitation, discharge flows) seems to vary a lot by time, location and so on (e.g. Figure 5). Regarding the economic submodel, detailed economic losses seem to be unknown (p. 9, top), so I believe that HAZUS and the input-output model were not in fact "calibrated" to the event. The authors could be clearer about this. Most*

*importantly, one would expect a summary regarding the authors' judgement of the OVERALL model performance in replicating the historical data.*

Yes, this is true. In short, we have confidence that the models are capturing the dominant physical mechanisms and are generally realistic. However, it is clear that there are issues with the representation of precipitation and hydrologic response. For this reason, we decided NOT to use the raw model output when doing the climate change simulations. The basic idea is that the models are not good enough to give us precise spatiotemporal values of the different variables (precipitation, streamflow etc.) However, we believe their representation of the dominant processes is good, and we trust that they are able to capture the CHANGES between the past and future – this is the reasoning behind the "Delta Method". We will explain this better.

The referee is right that the HAZUS model is not "calibrated" to the event in the sense that we can use an actual value of economic losses as a counterfactual to compare to the model results. The only data we can use to evaluate the model performance is the Department of Commerce estimated losses for the states of Washington and Oregon combined for this flooding event, which were approximately $1 billion dollars. In addition, the official building and inventory damages in Lewis county were estimated at $166 million. These are very close to our economic model results. In the revised manuscript, we will clearly state that the economic model is not calibrated and verified in the way that the physical models are.

We will incorporate a clearer statement regarding our assessment of the overall model.

*I can comment mainly on the economic aspects. The general idea of calculating direct losses first and then using an input-output model to calculate indirect or induced losses is plausible. The assumption that reconstruction is done by companies outside the affected area is also common. Regarding the obtained economic figures for the effect of climate change, they seem rather inconclusive. For example, what does it imply that physical damages of the considered event increase between 9 and 171% in Lewis County? The most relevant economic figure (for households, policy makers, insurance companies) would most likely be the expected annual losses and how these are affected by climate change. In particular, the probability of occurrence of the December 2007 event under present and future climate would be relevant in that regard. If we are talking about a 500-year event (as indicated on p. 2), future changes in this particular event would probably not be too relevant. Therefore, I wonder whether it would be possible to calculate hypothetical losses for, e.g. 20-, 50- and 100-year events. The meteorological records should provide the corresponding amounts of precipitation for these events and the economic losses could be obtained by using the model with the calibration for the December 2007 event. The PGW approach (as far as I understand it) would be applicable to those more frequent events analogously. Eventually, the expected annual losses (now and under climate change) could be calculated (see e.g. Velasco, 2015 for a simple approach).*

It is important to clarify that this is not a 500-year event for the entire basin. It was estimated to be a 500-year event only for the Doty stream gauge (we will make this clearer in the text).

The reviewer is correct, by focusing on such a large event, the future changes are probably not that relevant. In theory, repeating entire the methodology for a 20, 50, and 100-year event is possible – but it would take us a tremendous amount of time to complete the simulations that the reviewer is requesting (on the order of a year). In particular, obtaining the PGW simulations for the different return periods, and processing them through both hydrologic models and hydraulic model would take a very long time. However, the reviewer raises an important question, so we have attempted to address it using an alternative method, described below.

We started by analyzing the streamflow record for the Porter gauge (12031000, or gauge # 10 in the map). This gauge is downstream of the basin, and can capture the response of the full watershed. The gauge has a long record of data (70 years) so we can construct the flow duration curve and calculate the streamflow for different return periods (Figure 1). We then fit a log-normal distribution (black line) to the observed data. This is done to extrapolate to the 100-year return period (because we only have 70 years of data. We plot only the fitted values for the historical period in Figure 2, blue line.

[Figure]

Figure 1 Historical Annual Maximum Streamflow data for Porter gauge (green dots) and a fitted log-normal distribution (black lines).

[Figure]

*Figure 2 Fitted streamflow for different return periods for the historical period (blue) and the future (red). The changes in streamflow in the future are calculated by assuming a 15% increase in streamflow in the future. We then calculate the changes in return period.*

Our goal now is to evaluate how the streamflow would change in the future, without going through the exercise of the PGW-HEC-HMS/DHSVM-HEC-RAS. To do this, we begin by evaluating the change in streamflow that we calculated for the December 2007 event, which was between -1% to about 30% (Figure 12a of our manuscript). Hamlet and Lettenmaier (2007) evaluate how streamflow intensity has changed in the historical period. In their Figure 10 (copied below as Figure 3), they show how increased cool season precipitation variability results in flood risk for a 20-year, 50-year and 100-year return period over the Western US. Over our area of analysis, the changes have been on the order of 10% increase to 20% increase – and this holds for the three return periods. **Given these results, we believe that an increase in streamflow of around 15% in the future is a reasonable approximation.**

We then repeat the exercise of fitting the log-normal distribution, **but assume that the streamflow values in the future are 15% larger**. We then obtain a new log-normal fit for the future values of streamflow (Figure 2, red line). **So, while the 100-year event in the historical period was approximately 7700 cfs, this would correspond to an event with a 42-year return period in the future (see green dashed curve). The 50-year return period event will be a 23-year return period event in the future (dashed purple), while the 20-year event will be a 10-year event in the future (dashed black).**

[Figure]

**Figure 10.** Same as for Figure 7 except showing composites based on all years from 1973 to 2003 for the 20-year (left panels), 50-year (center panels), and 100-year (right panels) return intervals compared to the same values for the unconditional probability distributions (all years from pivot 2003 simulations).

*Figure 3 Figure from Hamlet and Lettenmaier 2007.*

Using these changes in return periods, we use a method similar to Velasco et al. 2015 to evaluate the losses for the historical and future events. Using HAZUS, we simulate flood events for 20-, 50- and 100-return periods, based on the default dataset from the software. We can then calculate the total economic losses for the three counties (Grays Harbor, Lewis and Thurston) using the same economic methodology as before:

| Return Period | Losses in Millions of Dollars |
|---|---|
| 20yr | $(14.75) |
| 50yr | $(17.12) |
| 100yr | $(23.42) |

Then, we calculate, the total losses for the historical period as the integral under the curve in Figure 4 ($920,000) and the total losses for the future as the integral under the orange curve ($1.2 million) for a total increase in losses of 23%.

| Return Period | 10 | 20 | 23 | 42 | 50 | 100 |
|---|---|---|---|---|---|---|
| Current | | $15 | | | $17 | $23 |
| Future | $15 | | $17 | $23 | | |

| Probability of Exceedance | 0.1 | 0.05 | 0.043 | 0.024 | 0.02 | 0.01 |
|---|---|---|---|---|---|---|

| Current | $0.92 | Average Annualized Losses (FEMA, Eq. 14-9) |
|---|---|---|
| Future | $1.12 | Average Annualized Losses (FEMA, Eq. 14-9) |
| **Change** | **23%** | |

[Figure]

*Figure 4 Economic Loss-Probability Curve for the Current and Future period*

We will incorporate these results in the conclusions section, as a method to extend the current results to other events of different return period. We realize this is a simplification, in the sense that we are assuming a 15% increase in streamflow for all return periods. However, this example nicely illustrates the methodology that one would follow to calculate the expected losses for different return periods. We would indeed like to do this in a future analysis, by

simulating a 20-year and 50-year return period event through the entire modeling system. Thank you for the interesting comment.

*Conclusion:*
*The overall quality of the paper is good and the suggested revision is somewhere between major and minor. The topic of the paper is relevant and the development of a coupled hydrologic, hydraulic and economic model is plausibly presented. The structure of the paper could be still improved and the implications of the results should be presented more clearly.*

We believe that the modifications to the manuscript that will be done to address your comments will significantly improve the structure of the paper and the implications of the results.

*Detailed aspects:*
*The abstract is very long (250 words). I would suggest leaving out the first three sentences, and starting the abstract with "In this work. . .".*

We understand the reviewer's concern and agree that that deleting these sentences would make the abstract more "to-the-point". However, after much deliberation (and consultation with other colleagues), we have decided to leave the first three sentences. The reason is that this manuscript is geared toward a wide audience (from atmospheric scientists to stakeholders) so it is important to put our work into context and motivate the research.

---

## Author Comment (AC2) · 13 Sep 2017

We thank Reviewer 2 for the thoughtful comments. We have transcribed only the comments that suggest modifications to the manuscript and responded to each comment below. We believe that these modifications will significantly improve the manuscript.

*Most of the analysis is well done. However, there are statements regarding the amount of water associated with the AR that is not backed up by any analysis. Are the authors confident the AR plume carried 1,500 kg/(ms) and the amount of precipitation that reached the basin was 70,000 CMS?*

Yes, the reviewer is correct. We will include a more detailed description of the calculations in our "Methods" section. We repeated the analysis using another dataset, and are now confident that the plume carried on the order of 1,500 kg/(ms) at its core. Figure 1a in our manuscript uses data from ERA-Interim Reanalysis, we have double checked with MERRA Reanalysis, and found similar results (see Figure 5a).

[Figure]

*Figure 1 (left) IVT from MERRA data for the December 2007 event. (right) IVT for the cross-section shown on the left.*

To calculate the water transported by this AR, we integrate IVT along the cross-section shown in Fig 5a. If we integrate along the full cross section, which has a length of ~1757 km, we get a value of 1.78e+09 kg/s (1.78e+6 CMS). If we only integrating along the core  (IVT >1500 kg/m/s) with a width of ~468 km, we get a value of 8.47e+08 kg/s (8.47e+5 CMS).  In the value reported in the text, we had actually only done it for the inner 400km, however, we will now report the more objective criteria for the core above 1500 kg/m/s. The value is now 84,700 $m^3$/s. However, it is important to keep in mind that this is the water carried by the AR – NOT the amount of precipitation. We will make this clear in the text.

*The descriptions of the models are well done. Error analysis of WRF precipitation is well explained, however hydrologic models calibration and verification may not be sufficiently presented.*

We will improve our description of the hydrologic models' calibration and verification, and update the manuscript as follows (the text in bold is new):

PG. 4 Line 28: In HEC-HMS, we partitioned the watershed into 64 sub-basins with homogenous soil and land cover properties based on data from SSURGO (USDA-NRCS) and NLCD 2011 (Homer et al., 2015). HEC-HMS provides the streamflow response of each of the sub-basins that drain to the Chehalis main channel. We calculated baseflow in three different ways: if there was a stream gauge, we used the USGS stream statistics; if the stream gauge was located downstream of a tributary, we calculated the initial base flow for the channel receiving from each sub-basin based on the fraction of the gauged area contributed by each sub-basin in the tributary; if there were no stream gauges available, we estimated the initial base flow through analogy with similar size sub-basins nearby. We used the Green and Ampt option in HEC-HMS to simulate infiltration in each sub-basin. **Given the limited observations, we estimated the Green and Ampt parameters (saturated hydraulic conductivity, effective porosity and wetting front suction head) based on the values reported in the literature for each hydraulic soil group.** For each sub-basin, we used the area-weighted properties. For purposes of calculating soil infiltration rates, we estimated percent impervious area using the land use and land cover maps obtained from SSURGO. The runoff transform uses the Soil Conservation Service (SCS) lag time. **The HEC-HMS simulated streamflow was compared to the observed streamflow at the USGS gauges listed in Table 1. The only parameter that was calibrated was the soil infiltration parameter which was adjusted within the range of each soil type. In addition, the final model setup with 64 sub-basins of homogeneous soil and land cover types was found to be the optimum representation of the basin, that resulted in streamflow closest to observations. If the basin is represented with fewer sub-basins, the HEC-HMS simulated streamflow does not capture the timing or magnitude of the peak in the observed hydrographs.**

For the DHSVM model:

Pg. 5 Line 16: **To calibrate DHSVM for the 2007 storm (12/01/2007~12/07/2017), we initially implemented a simple sensitivity analysis. DHSVM uses 18 different soil types, which the model links internally to soil hydraulic properties (e.g., saturated hydraulic conductivity, porosity, etc). We then determined sensitivity to the three dominant initial soil types (as suggested by Cuo et al. 2011), as well as other selected model parameters. We found that the soil maximum infiltration rate, and Manning's roughness coefficient (for channel flow) were the most sensitive parameters. We then developed a Monte Carlo simulation approach that randomly picked these parameters (between prescribed upper bound and lower bounds defined by Cuo et al. 2011). We compared simulated flows with USGS gauge station observed**

**streamflow (using RMSE) and identified the optimal parameter combinations within each sub-basin.**

*The HAZUS economic model is unclear with regard to the assumptions and uncertainties. The description of its setup, calibration and verification need to be further explained.*

It is important to emphasize that the HAZUS model is not "calibrated" to the event in the sense that we can use an actual value of economic losses as a counterfactual to compare to the model results. The only data we can use to evaluate the model performance is the Department of Commerce estimated losses for the states of Washington and Oregon combined for this flooding event, which were approximately $1 billion dollars. In addition, the official building and inventory damages in Lewis county were estimated at $166 million. These are very close to our economic model results. In the revised manuscript we will clearly state that the economic model is not calibrated and verified in the way that the physical models are.

We will improve our description of the HAZUS economic model assumptions and model setup. We will update the manuscript as follows (the text in bold is new):

Pg 4, Line 11: In terms of economic losses, we rely on HAZUS-MH 3.0 software with its standard infrastructure data and dasymetric dataset for buildings. **Using HAZUS-MH 3.0, we calculate the direct economic losses. However, to calculate their ripple effects throughout the economy – also called indirect losses – we rely on the Inventory-Dynamic Inoperability Input-Output Model (Inv-DIIM) proposed by Barker and Santos (2010) and on the 2008 input-output tables from IMPLAN (2015). The sector-specific inoperability levels and sector-specific recovery rates are calculated using the inventories of finished goods. Input-output data contain i**nformation about the trade flows across 16 different sectors that represent the economic structure of each of the counties within the state of Washington.

Pg. 6, Line 1: We calculated the direct economic losses using HAZUS (HAZard USa), a software developed by the Federal Emergency Management Agency (FEMA, 2015) to calculate economic losses associated to different natural disasters, including floods (see, among others, Ding et al. (2008), Banks et al. (2014), Gutenson et al. (2015)). We used HAZUS-MH version 3.0 and its default dasymetric datasets to calculate how the HEC-RAS-simulated flooding led to direct economic losses to agriculture (crops), to buildings and public infrastructure such as utilities and roads. The dasymetric data embedded in HAZUS, which includes information about the location and characteristics of the buildings (e.g. construction type, number of stories), allocates the use of land and of buildings by economic sectors so that one can estimate how the direct economic losses result in direct production capacity constraints and losses by sector. **Our HAZUS implementation contains several assumptions: as usual in the literature, production capacity constraints are based on the assumption of a homogeneous productivity per square**

foot for each industry in a specific county and on the assumption that industries operated at full capacity before the disaster. As a result, we set the production capacity constraints based on the pre-disaster total output by industry. When it comes to output in the agricultural sector, our assumption is to reduce it proportionally to the share of crop and livestock output in each county. We do not consider livestock losses independently as they are not reported by HAZUS.

Because each company or institution relies on a set of suppliers and purchasers to support its activities, they too will experience production losses as a result of the flood, even though they have not been flooded themselves. These indirect economic losses are estimated from the 2008 Input-Output tables extracted from IMPLAN at a 16-sector aggregation level (Avelino and Dall'erba, 2016). In addition to production losses, the combination of HAZUS and of input-output techniques allow us to quantify how local final demand decreases as a result of the employees suffering from labor income losses due to temporary closure of their workplace. **We assume that the expenditure structure remains fixed in the post-disaster period and that demand decreases proportionally to the decrease in income.** Reconstruction costs, on the other hand, correspond to a positive stimulus corresponding to the total repair costs of buildings, infrastructure, building content and vehicles that were destroyed or damaged during the flood. **While the first two elements lead to a construction stimuli, the last two support demand in manufacturing. Since IO models are based on producer prices and HAZUS provides repair costs in purchase prices, we assume that manufacturing orders include margins split 20/80% between transportation and trade.** Due to the small size of the economy of the affected counties, the model assumes that reconstruction efforts are supplied by companies located outside of the flooded area. The duration of the recovery phase is given by HAZUS (**Tables 14.1, 14.5 and 14.12 of FEMA, 2015**) and is assumed to be linear in time. The total economic impact in the three affected counties and the rest of Washington is then estimated using the Inventory-Dynamic Inoperability Input-Output Model (Inv-DIIM) of Barker and Santos (2010). It accounts for month-to-month cascading effects on production chains due to supply restrictions and existing inventories that mitigate some of these losses. In relation to other available input-output models, the Inv-DIIM offers a dynamic view of the inoperability and recovery processes, in addition to accounting for available inventories that can alleviate disruptions in the region (Avelino and Dall'erba, 2016). **The inventory data for the DIIM are based on the December 2007 inventory-to-sales ratio for manufacturing reported by the Federal Reserve Bank of St. Louis (2016). This approach has been suggested by Barker and Santos (2010). This ratio is 1.23 for the period under study and we choose to apply it homogeneously to all counties. Since the activities of wholesale and retail are recorded as margins, these sectors do not hold finished goods inventories. While they could hold "materials and supplies" and "work-in-progress" inventories, their data are not available.**

**References:**
**Federal Reserve Bank of St. Louis. Manufacturers: Inventories to Sales Ratio [Internet]. 2016. Available from: https://research.stlouisfed.org/fred2**

**Barker K, Santos JR. Measuring the efficacy of inventory with a dynamic input-output model. Int J Prod Econ. Elsevier; 2010;126(1):130–43.**

*The title is general with regard to the impacts of ARs and would be more clear if this was presented as a case study based on the December 2007 and RCP85 and RCP45 scenarios on Chehalis River Basin*

We understand the reviewer's concern, but would prefer to keep the title more general. Our reasoning is that we are presenting a tool that could be used in other places and for other events. While we focus on this one event to demonstrate how the tool works, we want to keep the focus of the paper on the tool itself. Following the suggestion of Reviewer 1, we will change the introduction and make it more general (without so much detail about the Dec 2007 event itself). This way, it will be clear that the paper is more related to the method than the case study.

---

## Author Response (AR1)

**Responses to Reviewer 1**

We thank Reviewer 1 for the thoughtful comments. We have responded to each comment below. We believe that these modifications will significantly improve the manuscript.

*Some parts of the introduction seem to go very much into detail. For example, the event is discussed with its detailed properties, but the region as such is only introduced in section 2. The introduction could be shorter and more general. All specific information could be moved to subsequent sections. For example, the concept of "pseudo-global warming" does not become clear from what is written in the introduction anyhow. The structure could be improved elsewhere. For example, climate change is the topic of section 2.3 as well as section 4. I suggest integrating section 2.3 into section 4.*

We agree with the Reviewer's comments. The details of the 2007 event were moved to Section 2 – Data and Methods – Page 3 Lines 19 – 31

On December 3rd 2007, an AR filamentary plume transporting more than 2,000 kg m$^{-1}$ s$^{-1}$ of water vapor at its core extended from the Tropical Pacific, west of Hawaii, to the coast of Oregon and Washington (Fig. 1a). Selecting the cross-section of the AR with most intense transport, and integrating IVT for all values exceeding 1,500 kg m$^{-1}$ s$^{-1}$ we can calculate the equivalent liquid water discharge. This AR carried approximately 847,000 m$^3$ s$^{-1}$ of liquid water across its inner core, or the equivalent of about 50 times the average discharge at the mouth of the Mississippi River. Temperatures rose 17°C in less than two hours ahead of the cold front (NOAA, 2008). Along this warm southwesterly tropical air mass, more than 70% of the water vapor and precipitation that reached the coast was of direct tropical origin (Eiras-Barca et al., 2017). The catastrophic flooding along the Chehalis River Basin was primarily due to unusually high and sustained hourly rainfall rates, concentrated in less than twenty-four hours, mainly on December 3. The conditions were exacerbated by warm air advection into the region by the AR, which produced rain on snow conditions and partially melted the existing shallow, low-elevation snow. Ten US Geological Survey (USGS) stream gauges experienced record flooding, including four on the Chehalis River or its tributaries (Grand Mound, Porter, Doty and South Fork Chehalis; see Fig. 1b for station locations). The peak discharge measured at Doty was a 500 year event - the only 500 year stream peak event ever recorded in Western Washington.

The sentences that provide details about the "pseudo-global warming" method was shifted to Section 2.3 – Climate Change Simulations Page 7 Lines 13-25.

**2.3   Methods: Climate Change Simulations**

To understand how the December 2007 event would change if it occurred in a warmer climate, we used a "pseudo-global warm-ing" (PGW) approach (Schär et al., 1996; Sato et al., 2007; Kawase et al., 2009; Lynn and Druyan, 2009; Rasmussen et al., 2011; Lackmann, 2013, 2015). The PGW can provide complementary information to the traditional downscaling approach as it gives more physical insight into detailed spatial processes and potentially a better way of communicating with regional stakeholders, as argued by Hazeleger et al. (2015). In this approach, the lateral and initial boundary conditions used in the

We have decided not to integrate section 2.3 into 4. The reason is that section 2.3 is in the "Methods" section, where we go into the details of the PGW methodology. However, section 4 is part of results, so we think it would be confusing for the reader.

We changed the titles of each section to clarify each section:
 1 Introduction
2 Data and Methods
       2.1 Data: Observations
       2.2 Methods: Models
       2.3 Methods: Climate Change simulations
          'Delta Method' for Model Simulations goes here but is not labeled as a separate
          section.
3 Results: Historical Simulations
4 Results: Climate Change Simulations
5 Conclusions

*The third section considers the simulation of the actual event, or the model calibration, as I would name it. However, it remains a bit unclear how well the overall model fits the observed data. The fit of some submodels (for precipitation, discharge flows) seems to vary a lot by time, location and so on (e.g. Figure 5). Regarding the economic submodel, detailed economic losses seem to be unknown (p. 9, top), so I believe that HAZUS and the input-output model were not in fact "calibrated" to the event. The authors could be clearer about this. Most importantly, one would expect a summary regarding the authors' judgement of the OVERALL model performance in replicating the historical data.*

We have added a new paragraph in Pg 10 lines 3-11.

Overall we find that individually the models of the integrated system realistically capture the dominant physical/economic processes. However, it is clear that there are problems with some variables, particularly precipitation and the associated hydrologic response. For this reason, we decided not to use the raw model output (from WRF, HEC-HMS, DHSVM or HEC-RAS) to drive the subsequent model in the historical simulations (Fig. 2). The Instead,we use the individual historical model simulationsare used as the 'control' to compare with , forced with observations. To simulate the climate change simulations, described below. response, the observations are then multiplied by a factor that accounts for the changes projected by the models in a warmer climate (as described in Section 2.3). The underlying idea is that the models cannot provide precise

spatiotemporal values of the different variables; however, because their representation of the dominant processes is realistic, we trust they are able to capture the changes between the past and the future. This is the rationale behind the 'delta-method'.

The referee is right that the HAZUS model is not "calibrated" to the event in the sense that we can use an actual value of economic losses as a counterfactual to compare to the model results. We added the following sentence Pg 9 Line 33:

It is important to  clarify that we do not have a counterfactual that can be used to calibrate the economic model in the same way that we calibrate the physical models.

*I can comment mainly on the economic aspects. The general idea of calculating direct losses first and then using an input-output model to calculate indirect or induced losses is plausible. The assumption that reconstruction is done by companies outside the affected area is also common. Regarding the obtained economic figures for the effect of climate change, they seem rather inconclusive. For example, what does it imply that physical damages of the considered event increase between 9 and 171% in Lewis County? The most relevant economic figure (for households, policy makers, insurance companies) would most likely be the expected annual losses and how these are affected by climate change. In particular, the probability of occurrence of the December 2007 event under present and future climate would be relevant in that regard. If we are talking about a 500-year event (as indicated on p. 2), future changes in this particular event would probably not be too relevant. Therefore, I wonder whether it would be possible to calculate hypothetical losses for, e.g. 20-, 50- and 100-year events. The meteorological records should provide the corresponding amounts of precipitation for these events and the economic losses could be obtained by using the model with the calibration for the December 2007 event. The PGW approach (as far as I understand it) would be applicable to those more frequent events analogously. Eventually, the expected annual losses (now and under climate change) could be calculated (see e.g. Velasco, 2015 for a simple approach).*

We have added a new section called "Interpretation". Pg. 11 Lines 22- Pg 12 Line 19. And a new figure 13. This incorporates the new analysis of the expected annual losses.

**Interpretation**

Despite the fact that some sub-basins experience lower streamflow in the climate change simulation (see Skookumchuck and Newaukum in Fig. 10), streamflow throughout the main stem of the Chehalis increased. This implies that the dramatic increases in flooding of the headwaters (see Doty in Fig. 10) dominated the system response and caused flooding in populated downstream areas along the main stem of the river including Centralia and Chehalis (the largest population centers in the basin). Our results highlight that the economic impacts are very sensitive to the geographical location of inundated area and depth. The parts of the basin with large population centers are most vulnerable to direct economic losses and account for most of the stock damages (Table 3). But this is not the only factor. Indeed, Thurston County has strong trade linkages to other regions (such as the Seattle metropolitan area) and for this reason, despite modest changes in direct impacts, the net impact on trade increased significantly in the climate change simulation (480-619%) (Table 3). This indicates that, depending on the hydrological impacts, the simulated economic scenarios can lead to flooding patterns that impact key interconnected sectors of the economy, significantly increasing negative spillover effects.

Interestingly, despite general increases in streamflow in the climate change simulation, the  changes in inundation extent are minimal (Fig. 12b). The reason for this is that the December 2007 event was so large that the flooding extended throughout much of the flood plain to the bounding and steeper hills. As a result, the  changes in economic impacts might not be very large, for an event of such low probability of exceedance. Smaller events may well be (proportionately) more affected under climate change in this river basin (clearly, the extent of the flood plain and characteristics of the bounding topography are basin-specific). We were able to get some insight into the nature of the basin's response to changes in more modest floods using a simplified method (in contrast to the full chain of model calculations that underlie our estimates for the 2007 flood) by using the default data for flood extent and depth for different return periods from HAZUS (without performing the atmospheric, hydrologic or hydraulic analysis) and applying the changes to gauge observations.

The Porter stream gauge (gauge 10 in Fig. 1) provides representative data for the entire watershed, and allows us to identify the streamflow for different return periods (Fig 13). Assuming that climate change will result in 15% more streamflow for all return periods (an assumption based on our PGW results and results from Hamlet and Lettenmaier (2007); see their Fig.10), we used HAZUS, and a method similar to Velasco et al. (2015) to evaluate the losses for historical and future events. We then calculated expected total losses for the historical period as the integral under the blue curve in (Fig 13c) ($920,000) and expected total losses for the changed climate condition as the integral under the red curve ($1.2 million) for a total increase in expected losses of 23%. In the future, we plan to repeat this analysis using the full integrated model chain to obtain more realistic values for the changes in streamflow, which would replace the assumed 15% increase in streamflow independent of return period.

[Figure]

**Figure 13.** Methodology to calculate the historical and future expected annual losses using only HAZUS and streamflow observations a) Flow Duration curve for the Porter gauge and the fitted log-normal distribution. b) Fitted streamflow for different return periods for the historical period (blue) and the future (red). The changes in streamflow in the future are calculated by assuming a 15% increase in streamflow in the future. We then calculate the changes in return period c) Economic Loss-Probability curve for the current and future period.

*Conclusion:*
*The overall quality of the paper is good and the suggested revision is somewhere between major and minor. The topic of the paper is relevant and the development of a coupled hydrologic, hydraulic and economic model is plausibly presented. The structure of the paper could be still improved and the implications of the results should be presented more clearly.*

We believe that the modifications to the manuscript to address your comments significantly improved the structure of the paper and the implications of the results.

*Detailed aspects:*
*The abstract is very long (250 words). I would suggest leaving out the first three sentences, and starting the abstract with "In this work. . .".*

We understand the reviewer's concern and agree that that deleting these sentences would make the abstract more "to-the-point". However, after much deliberation (and consultation with other colleagues), we have decided to leave the first three sentences. The reason is that this manuscript is geared toward a wide audience (from atmospheric scientists to stakeholders) so it is important to put our work into context and motivate the research.

*Responses to Reviewer 2*

We thank Reviewer 2 for the thoughtful comments. We have transcribed only the comments that suggest modifications to the manuscript and responded to each comment below. We believe that these modifications significantly improved the manuscript.

*Most of the analysis is well done. However, there are statements regarding the amount of water associated with the AR that is not backed up by any analysis. Are the authors confident the AR plume carried 1,500 kg/(ms) and the amount of precipitation that reached the basin was 70,000 CMS?*

Yes, the reviewer is correct. We included a more detailed description of the calculations in our "Methods" section. Pg. 3, lines 19-23.

On December 3rd 2007, an AR filamentary plume transporting more than 2,000 kg m$^{-1}$ s$^{-1}$ of water vapor at its core extended from the Tropical Pacific, west of Hawaii, to the coast of Oregon and Washington (Fig. 1a). Selecting the cross-section of the AR with most intense transport, and integrating IVT for all values exceeding 1,500 kg m$^{-1}$ s$^{-1}$ we can calculate the equivalent liquid water discharge. This AR carried approximately 847,000 m$^3$ s$^{-1}$ of liquid water across its inner core, or the equivalent of about 50 times the average discharge at the mouth of the Mississippi River. Temperatures rose 17°C in less than

[Figure]

Figure 1 (left) IVT from MERRA data for the December 2007 event. (right) IVT for the cross-section shown on the left.

*The descriptions of the models are well done. Error analysis of WRF precipitation is well explained, however hydrologic models calibration and verification may not be sufficiently presented.*

HEC-HMS on Page 5, lines 1-11

basin, and calculated the . Given the limited observations, we estimated the Green and Ampt parameters (saturated hydraulic conductivity, effective porosity and wetting front suction head) based on the values reported in the literature for each hydraulic soil group. For each sub-basin, we used the area-weighted properties. For purposes of calculating soil infiltration rates, we estimated percent impervious area using the land use and land cover maps obtained from SSURGO. The runoff transform uses the Soil Conservation Service (SCS) lag time. The HEC-HMS simulated streamflow was compared to the observed streamflow at the USGS gauges listed in Table 1. The only parameter that was calibrated was the soil infiltration parameter which was adjusted within the range of each soil type. In addition, the final model setup with 64 sub-basins of homogeneous soil and land cover types was found to be the optimum representation of the basin and it resulted in streamflow closest to observations. If the basin is represented with fewer sub-basins, the HEC-HMS simulated streamflow does not capture the timing or magnitude of the peak in the observed hydrographs.

For the DHSVM model Pg. 5 Lines 23-30

gauges. We calibrated the maximum infiltration rates for each soil type and (for DHSVM)using To calibrate DHSVM for the 2007 storm we initially implemented a simple sensitivity analysis. DHSVM uses 18 different soil types which the model links internally to soil hydraulic properties (e.g., saturated hydraulic conductivity, porosity, etc). We then determined sensitivity to the three dominant initial soil types (as suggested by Cuo et al., 2011), as well as other selected model parameters. We found that the soil maximum infiltration rate, and Manning's coefficients for each channel reach. roughness coefficient (for channel flow) were the most sensitive parameters. We then developed a Monte Carlo simulation approach that randomly picked these parameters (between prescribed upper bound and lower bounds defined by Cuo et al., 2011). We compared simulated flows with USGS gauge station observed streamflow (using RMSE) and identified the optimal parameter combinations within each sub-basin.

*The HAZUS economic model is unclear with regard to the assumptions and uncertainties. The description of its setup, calibration and verification need to be further explained.*

It is important to emphasize that the HAZUS model is not "calibrated" to the event in the sense that we can use an actual value of economic losses as a counterfactual to compare to the model results. The only data we can use to evaluate the model performance is the Department of Commerce estimated losses for the states of Washington and Oregon combined for this flooding event, which were approximately $1 billion dollars. In addition, the official building and inventory damages in Lewis county were estimated at $166 million. These are very close to our economic model results. In the revised manuscript we clearly state that the economic model is not calibrated and verified in the way that the physical models are. Pg. 9 Line 33

It is important to re-iterate at this point clarify that we do not have a counterfactual that can be used to calibrate the economic model in the same way that we calibrate the physical models.

We improved our description of the HAZUS economic model assumptions and model setup. We updated the manuscript as follows:

Pg. 4 Line 9:

and Forecast (WRF) atmospheric model simulations. In terms of direct economic losses, we rely on  infrastructure data and dasymetric dataset for buildings  which is embedded in the standard release of HAZUS-MH 3.0. To calculate their ripple effects throughout the local supply chain (also called indirect losses) we rely on the 2008 input-output tables  from IMPLAN (2015). The sector-specific inoperability levels and sector-specific recovery rates are calculated using the inventories of finished goods. Input-output data contain information about trade flows across 16 different sectors that represent the economic structure of each of the counties within the state of Washington. They were obtained from IMPLAN (2015).

Pg. 6, Line 19 – Pg 7 Line 11

Our HAZUS implementation contains several assumptions: as usual in the literature, production capacity constraints are based on the assumption of a homogeneous productivity per square foot for each industry in a specific county and on the assumption that industries operated at full capacity before the disaster. As a result, we set the production capacity constraints based on the pre-disaster total output by industry. While HAZUS is able to calculate damages to crop and some crop areas were flooded during the event, crop losses are null because the event took place several months before the planting season. Buildings located on farmland were damaged however, and their repair or reconstruction costs follow the same methodology as similar costs, as described further below.

Because each company or institution relies on a set of suppliers and purchasers to support its activities, they too will experience production losses as a result of the flood  even though they have not been flooded themselves. These indirect economic losses are estimated from the 2008 Input-Output tables extracted from IMPLAN at a 16-sector aggregation level (Avelino and Dall'erba, 2016). In addition to production losses, the combination of HAZUS and of input-output techniques allow us to quantify how local final demand decreases as a result of the employees suffering from labor income losses due to temporary closure of their workplace. We assume that the expenditure structure remains fixed in the post-disaster period and that demand decreases proportionally to the decrease in income. Reconstruction costs, on the other hand, correspond to a positive stimulus  encompassing the total repair costs of buildings, infrastructure and vehicles that were destroyed or damaged during the flood. Since IO models are based on producer prices and HAZUS provides repair costs in purchase prices, we assume that manufacturing orders include margins split 20/80% between transportation and trade. Due to the small size of the economy of the affected counties, the model assumes that reconstruction efforts are supplied by companies located outside of the flooded area. The duration of the recovery phase is given by HAZUS  (Tables 14.1, 14.5 and 14.12 of FEMA, 2015) and assumed to be linear in time. The total economic impact in the three affected counties and the rest of Washington is then estimated using the Inventory-Dynamic Inoperability Input-Output Model (Inv-DIIM) proposed by Barker and Santos (2010). In relation to other available input-output models, the Inv-DIIM offers a dynamic view of the inoperability and recovery processes, in addition to accounting for available inventories that can alleviate disruptions in the region (Avelino and Dall'erba, 2016). The inventory data for the DIIM are based on the December 2007 inventory-to-sales ratio for manufacturing reported by the Federal Reserve Bank of St. Louis in 2016. This ratio has been suggested by Barker and Santos (2010) and is equivalent to 1.23 for the period under study. We apply it homogeneously to all counties. Since the activities of wholesale and retail are recorded as margins, these sectors do not hold finished goods inventories.

*The title is general with regard to the impacts of ARs and would be more clear if this was presented as a case study based on the December 2007 and RCP85 and RCP45 scenarios on Chehalis River Basin*

We understand the reviewer's concern, but would prefer to keep the title more general. Our reasoning is that we are presenting a tool that could be used in other places and for other events. While we focus on this one event to demonstrate how the tool works, we want to keep the focus of the paper on the tool itself. Following the suggestion of Reviewer 1, we changed the introduction and make it more general (without so much detail about the Dec 2007 event itself). This way, it will be clear that the paper is more related to the method than the case study.

[revised manuscript text omitted]

**Observed (Livneh)**
**Average Precipitation Dec 1-4**

**WRF-Simulated**
**Average Precipitation Dec 1-4**

[Figure]

**WRF Precipitation Bias**
**Chehalis River Basin**

[Figure]

**Figure 4.** (a) Observed daily precipitation (mm day$^{-1}$) averaged for Dec 1-4 of 2007 from Livneh et al. (2013), b) WRF-control simulated precipitation for the same period and c) bias in simulated precipitation for each of the HEC-HMS sub-basins within the Chehalis basin.

[Figure]

**Figure 5.** USGS observed (solid black), HEC-HMS simulated control (dashed blue) and DHSVM simulated (dotted blue) discharge for 4 representative sub-basins within the Chehalis.

[Figure]

**Figure 6.** (a., c. and d.) USGS observed (solid) and simulated (dashed) stage for three cross-sections of the Chehalis river main stem as represented by HEC-RAS. b. Flood extent and depth map as simulated by HEC-RAS.

[Figure]

**Figure 7.** a) WRF-control simulated IVT (kg m-1 s-1) for December 3, 2007, b) WRF-PGW simulated IVT (kg m-1 s-1) for December 3, 2007 for the upper scenario, c) Absolute change in IVT, between WRF-PGW upper scenario and WRF-control.

[Figure]

**Figure 8.** (Changes between WRF-PGW for the upper scenario and WRF-control (inner WRF domain), averaged for the Dec 1-4 period for a) water vapor mixing ratio percent change at 800mb, b) relative humidty % change at 800mb, c) absolute change in cloud water mixing ratio at 800mb and d) % change in precipitation e) % precipitation change area averaged over all Chehalis sub-basins of the HEC-HMS model.

[Figure]

**Figure 9.** (a) % change in precipitation for the lower (green) and upper (red) scenarios, as simulated by WRF for all Chehalis sub-basins used in the HEC-HMS simulations. (b) % change in streamflow for the lower (green) and upper (red) scenarios, as simulated by HEC-HMS for all Chehalis sub-basins

[Figure]

**Figure 10.** Streamflow hydrographs for HEC-HMS-PGW upper scenario (dashed) and HEC-HMS control (solid) for select sub-basins in the Chehalis.

[Figure]

**Figure 11.** Streamflow hydrographs for observed (black) and simulated using HEC-HMS (dashed) and DHSVM (dotted) for the lower scenario (green) and higher scenario (red).

[Figure]

**Figure 12.** Change in streamflow and flood depth along main channel.

[Figure]

**Figure 13.** Methodology to calculate the historical and future expected annual losses using only HAZUS and streamflow observations a) Flow Duration curve for the Porter gauge and the fitted log-normal distribution. b) Fitted streamflow for different return periods for the historical period (blue) and the future (red). The changes in streamflow in the future are calculated by assuming a 15% increase in streamflow in the future. We then calculate the changes in return period c) Economic Loss-Probability curve for the current and future period.

---

## Author Response (AR2)

*Response to Anonymous Referee #1 (Report 2)*

*The authors have convincingly addressed the comments from the first review round.*

*Regarding mainly my field of expertise, the economic part, the authors provide additional figures that greatly improve the understandability of the economic results.*
*A more detailed explanation and interpretation of the HAZUS model is added. Although the use of such an "external" model still has (and always will have) some sort of "black box"-character, I agree that the use of the HAZUS model seems appropriate in the present case. The authors now emphasize that the model is not really calibrated and that overall damages estimates were used as reference.*
*The (newly) presented underlying economic assumptions (industries at full a capacity prior to the event, fixed expenditure structure, potential positive impacts, homogeneous productivity per square foot, ...) are reasonable and indeed standard in similar economic models.*
*The calculation of the loss-probability curves and the corresponding expected annual losses in Figure 13 c) constitutes the major improvement regarding the presentation of the economic results.*

*Unfortunately, a simulation does not seem to exist for the 1-in-100 years event under the +15% streamflow scenario. However, one can reasonably assume that the losses would be at least as high as for the present 1-in-100 years event (probably much higher).*
*Thus, if the expected losses are really calculated as the area under the curves as marked in the figure, the expected losses for the future scenario are too low.*
*Furthermore, by looking at the figure, it seems that the marked area under the blue curve is actually larger than the one under the red curve (because of the missing 1-in-100 years event). Taking the unavailability of results for the 1-in-100 years event under the +15% streamflow scenario as given, I would suggest using the the losses from the present 1-in-100 years event to calculate a lower bound for the corresponding future event, which would lead to a more realistic (and higher) expected value than the presented value (1.2 million $).*

*Subsequently, the authors have done a good job in revising the paper. I would leave the remaining point regarding the expected losses (Figure 13c) to the authors to deal with in a potential editorial process and I would not demand another resubmission for this small point.*
*In other words, I think the paper should be "accepted subject to (very) minor revisions"*

We thank the reviewer for the constructive comments. We realized that Figure 13 could be significantly improved if we performed the 1-in-100 years event under the +15% scenario, so we requested additional time to perform these calculations to obtain a more comprehensive loss curve for the historical and future periods. We have modified Figure 13b and 13c to incorporate the Reviewer's suggestions. The manuscript has been modified as follows:

Pg 12, Line 14: "We then calculated expected total losses for the historical period as the integral under the blue curve in (Fig 13c) ($6.2 million) and expected total losses for the changed climate condition as the integral under the red curve ($8.6 million) for a total increase in expected losses of 39%. In the future, we plan to repeat this analysis using the full integrated model chain to obtain more realistic values for the changes in streamflow, which would replace the assumed 15% increase in streamflow independent of return period."

[Figure]